# Design of bacteriophage T4-based artificial viral vectors for human genome remodeling

Jingen Zhu[1], Himanshu Batra[1,3], Neeti Ananthaswamy[1,3], Marthandan Mahalingam[1,3], Pan Tao[1,3], Xiaorong Wu[1], Wenzheng Guo[1], Andrei Fokine[2] & Venigalla B. Rao[1] ✉

Designing artificial viral vectors (AVVs) programmed with biomolecules that can enter human cells and carry out molecular repairs will have broad applications. Here, we describe an assembly-line approach to build AVVs by engineering the well-characterized structural components of bacteriophage T4. Starting with a 120 × 86 nm capsid shell that can accommodate 171-Kbp DNA and thousands of protein copies, various combinations of biomolecules, including DNAs, proteins, RNAs, and ribonucleoproteins, are externally and internally incorporated. The nanoparticles are then coated with cationic lipid to enable efficient entry into human cells. As proof of concept, we assemble a series of AVVs designed to deliver full-length *dystrophin* gene or perform various molecular operations to remodel human genome, including genome editing, gene recombination, gene replacement, gene expression, and gene silencing. These large capacity, customizable, multiplex, and all-in-one phage-based AVVs represent an additional category of nanomaterial that could potentially transform gene therapies and personalized medicine.

Viruses are the most abundant and widespread organisms on Earth. They are also some of the most efficient biological machines[1,2]. Despite their small size and simple genetic makeup, viruses can cause deadly infections and global pandemics, such as AIDS, Flu, and COVID-19. This is because viruses evolved efficient mechanisms to replicate and assemble progeny in fast timescales, on the order of minutes in the case of bacterial viruses (bacteriophages or simply phages)[3,4]. If some of the efficient viral mechanisms could be harnessed by building artificial viral vectors (AVVs), programmed with therapeutic molecules, such viruses, instead of replicating in the host, could perform beneficial repairs to restore human health. Such AVVs could potentially replace defective genes, produce therapeutic molecules, kill cancer cells, and so on[5–10]. Despite many attempts over the years[6,11], the development of AVVs remained at an early stage.

Natural human viruses, adeno-associated viruses (AAVs) with ~5 Kbp size single-stranded DNA genome and lentiviruses with ~10 Kbp size single-stranded RNA genome, have been engineered to deliver

therapeutic DNA or RNA as part of their genome[12–14]. However, these viral vectors have limitations. They can at best deliver one or two therapeutic genes, and pose difficulties to incorporate additional therapeutic molecules essential for complex repairs. Safety concerns such as broad infectivity to human cells, pre-existing immunity, and potential integration into the host genome are additional serious issues[14,15].

Here, we describe an AVV platform using phage T4. T4 belongs to *Straboviridae* family and infects *Escherichia coli* bacterium[16,17]. With an infection efficiency nearing 100%[18], and replicating at a rate of ~20–30 min per cycle[19], T4 is one of the most efficient viruses known. It contains a large 120 × 86 nm prolate icosahedral capsid (head) assembled with 930 molecules or 155 hexameric capsomers of the major capsid protein gp23* (* represents the cleaved mature form), 55 copies or 11 pentamers of gp24* at eleven of the twelve vertices, and 12 copies of the portal protein gp20 at the unique twelfth vertex (Fig. 1a–c)[20–22]. The portal vertex is a ring structure with a ~35 Å central channel through which the viral genome is transported into capsid by

[1]Bacteriophage Medical Research Center, Department of Biology, The Catholic University of America, Washington, DC 20064, USA. [2]Department of Biological Sciences, Purdue University, West Lafayette, IN 47907, USA. [3]These authors contributed equally: Himanshu Batra, Neeti Ananthaswamy, Marthandan Mahalingam, Pan Tao. ✉e-mail: rao@cua.edu

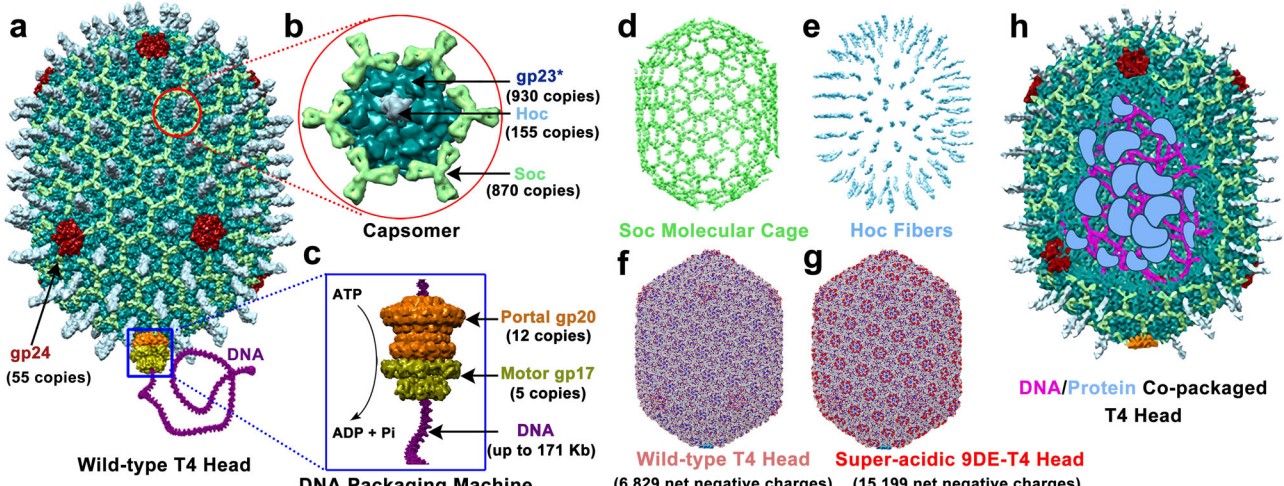

**Fig. 1 | Structural components for assembly of bacteriophage T4-AVVs.**
**a** Structural model of phage T4 head (capsid)[44]. Pentameric gp24 vertices are shown in red. **b** Enlarged capsomer shows the hexameric arrangement of major capsid protein gp23 (dark green), Soc trimers (light green), and Hoc fiber (cyan)[44]. **c** Enlarged DNA packaging machine structural model comprised of gp20 portal dodecamer (PDB 3JA7) (brown) and pentameric gp17 DNA packaging motor (PDB 3CPE) (yellow)[24,44]. **d** Eight hundred and seventy Soc molecules assembled at the quasi-three-fold axes form a molecular cage around T4 capsid[21] (PDB 5VF3). **e** One hundred and fifty-five Hoc fibers emanate from the centers of capsomers[34] (PDB 3SHS). **f, g** Molecular surfaces of wild-type (WT) T4 capsid[22] (3.4 Å, PDB 7VS5) (**f**) and super-acidic 9DE-T4 capsid (3.9 Å) (**g**) are colored according to electrostatic potential. The color ranges from red, corresponding to a potential of −5 kT/e⁻, to blue, corresponding to a potential of +5 kT/e⁻. The WT-T4 capsid has 6,829 net negative charges and the 9DE-T4 capsid has 15,199 net negative charges.
**h** Schematic of head packaged with foreign proteins and DNAs in its interior space.

an ATP-powered pentameric molecular motor attached to it (Fig. 1c)[23–25]. After one headful of genome, equivalent to ~171 Kbp linear dsDNA, is packaged[26,27], the motor dissociates and neck proteins assemble followed by tail and tail fiber assembly to generate an infectious virion[28–31].

The surface of T4 capsid is arrayed with two nonessential outer capsid proteins, Soc (*small outer capsid protein*) (9.1 kDa; 870 copies per capsid) and Hoc (*highly antigenic outer capsid protein*) (40.4 kDa; 155 copies per capsid) (Fig. 1b, d, e)[20,32]. Soc, a tadpole-shaped molecule, binds as a trimer at the quasi-three-fold axes. Each Soc subunit acts as a molecular clamp, clasping two adjacent capsomers. These 870 clamps form a molecular cage around the capsid (Fig. 1d), greatly reinforcing the capsid that is pressurized by tightly packed DNA approaching crystalline density[33]. Hoc on the other hand is a ~185 Å-long fiber composed of four Ig-like domains, with the C-terminal domain bound to the center of each gp23 capsomer. The 155 symmetrically positioned Hoc fibers emanate from T4 head (Fig. 1e)[34]. Unlike Soc, Hoc provides only marginal stability to capsid. Its main function might be to allow phage to adhere to bacterial host or mammalian mucosal surfaces through its Ig-like domains[34–36].

There are many reasons why T4 is an ideal platform to build AVVs, a concept that evolved over our >40 years of genetic, biochemical, and structural analyses. First, the architecture of T4 phage with a stable capsid and external surface exposing 1,025 nonessential molecules, and an internal volume that can accommodate up to ~171 Kbp DNA and ~1,000 molecules of internal proteins (IPs), provide ample cargo space to incorporate therapeutic biomolecules[21,37–40]. Second, there is extensive knowledge of the genetic and biochemical mechanisms of head assembly and genome packaging, enabling in vitro manipulations to build AVVs in a test tube[25,26,41–43]. Third, we have determined the atomic structures of almost all the capsid and packaging motor components, providing valuable information to engineer the T4 nanoparticle[21–24,33,34,44]. Fourth, Soc and Hoc can serve as efficient adapters to tether foreign proteins to the exterior of T4 capsid[37,45,46]. Both have nanomolar affinity and exquisite specificity to T4 capsid, which are crucial for in vitro assembly[47,48]. In parallel, Black and co-workers have developed genetic strategies to package foreign proteins, such as Cre recombinase, within the capsid[49,50]. Fifth, a robust in vitro

DNA packaging system has been developed, allowing an emptied T4 capsid to be re-filled with foreign DNA using the powerful DNA packaging motor[51–53]. Finally, a T4 CRISPR engineering strategy has been established, which facilitates the insertion of foreign DNA fragments into the phage genome, generating recombinant phages with unique phenotypic properties[38,54–59].

These provide an extraordinary foundation to design an AVV platform using the T4 phage. We develop an assembly-line approach, beginning with an empty capsid shell containing only three essential capsid proteins, gp23*, gp24*, and gp20. Layers of cargo molecules, including DNAs, proteins, RNAs, and their complexes, are incorporated into both inside and outside of the shell by a sequential assembly process. The negatively charged capsids (Fig. 1f, g), are then coated with positively-charged lipid molecules to mimic an envelope around these virus-like nanoparticles. The assembled artificial viral particles mimic natural viruses with a lipid coat, surface-exposed molecules, capsid shell, and packaged "genome" and proteins (Fig. 1h).

Here, as proof of concept, a series of T4-AVVs are assembled containing combinations of payloads to remodel the human genome in cultured cells. These include genome editing, gene recombination, gene replacement, gene expression, and gene silencing. For example, in one configuration, an AVV is programmed with five different components; Cas9 genome editing nuclease, Cre recombinase, two gRNAs, donor DNA, and reporter plasmids. Furthermore, we demonstrate delivery and expression of ~17 Kbp polygene consisting of full-length human dystrophin gene fused in tandem with three reporter genes. Such a large capacity, all-in-one, multiplex, programmable, and phage-based AVVs represent a distinct category of nanomaterial that could be used in the future for a variety of gene therapies and personalized medicine. To our knowledge, this is the first report on designing such lipid-coated phage AVVs that, considering the abundance of phage nanostructures in nature, would open new avenues for creating novel delivery vehicles.

## Results

### Assembly of T4 artificial viral vectors
T4-AVVs were assembled by sequential incorporation of purified biomaterials to generate a virus structural mimic (Fig. 2a and

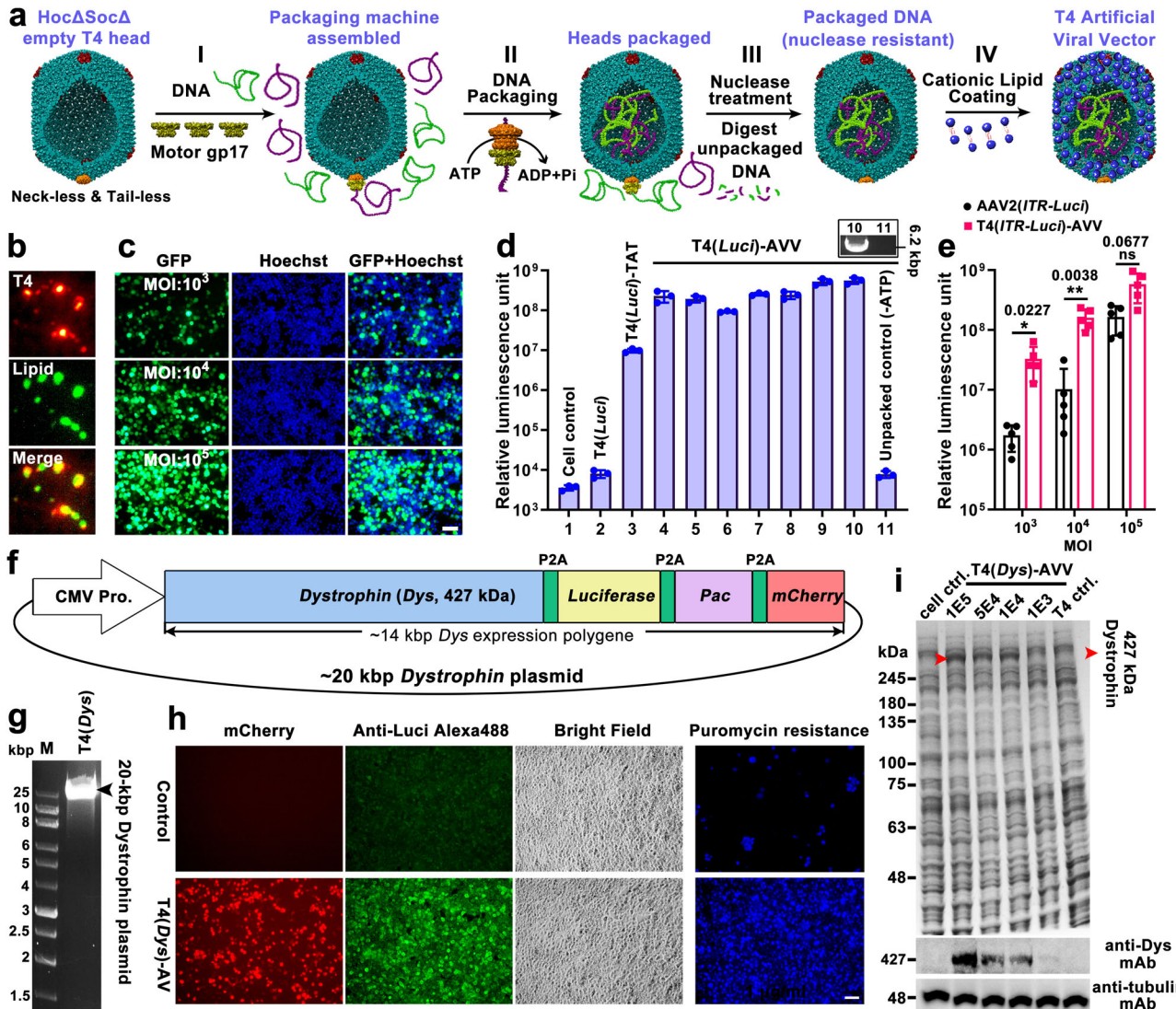

**Fig. 2 | T4-AVVs efficiently deliver genetic payloads into human cells.**
**a** Schematic of sequential assembly of DNA-packaged T4-AVVs. **b** Alexa Fluor 594 (red) labeled T4 capsid colocalized with nitrobenzoxadiazole (NBD, green) labeled cationic lipid molecules. **c** T4(*GFP*)-AVVs efficiently delivered packaged *GFP* DNA into 293 T cells, as determined by GFP expression at different MOIs (multiplicity of infection, ratio of AVV particles to cells). Cell nuclei were stained with Hoechst. Bar = 50 μm. **d** Transduction efficiencies of T4(*Luci*)-AVVs coated with different cationic lipids, as determined by luciferase expression. 1, cell control (no treatment); 2, T4(*Luci*) control (no lipid coating and no TAT); 3, T4(*Luci*)-TAT control (TAT-displayed, no lipid); 4-10, T4(*Luci*)-AVVs coated with various lipids: 4, LPF3K-AVVs; 5, LPFLTX-AVVs; 6, LPFStem-AVVs; 7, EXPI-AVVs; 8, FECT-AVVs; 9, LPFRNAiMAX-AVVs; 10, LPF2K-AVVs; 11, unpackaged control (same as #10 but no ATP). The top right box shows the packaged *Luci* DNA in groups 10 and 11. Values represent mean with standard deviation (SD) (*n* = 3). **e** Transduction efficiencies of T4(*ITR-Luci*)-AVV and single-stranded AAV2(*ITR-Luci*) at a MOI of $10^3$, $10^4$, or $10^5$. The T4-packaged *ITR-*

*Luci* plasmid (*AAV2ITR-CMV enhancer and promoter-fireflyLuci-hGH polyA*) has the same sequence as the one packaged into AAV2 particles. Values represent mean with SD (*n* = 5). *$P < 0.05$, **$P < 0.01$, and ns, not significant. Paired *t* test (two-tailed) was used for comparison in each MOI. **f** Schematic of ~20 kbp *dystrophin* (*Dys*) plasmid. **g** Agarose gel electrophoresis showing T4-packaged *Dys* plasmid (~2.5 molecules per head). **h** mCherry expression, *luciferase* expression, and puromycin resistance (*Pac* expression) in 293 T cells transduced by T4(*Dys*)-AVVs. *Luci* expression was detected by cellular immunofluorescence using Alexa488-labeled anti-Luci antibody. Puromycin-resistant 293 T cells following transduction with T4(*Dys*)-AVVs were stained by Hoechst, while the sensitive cells floated and were washed off. Bar = 50 μm. **i** *Dystrophin* expression in 293T cells transduced by T4(*Dys*)-AVVs. Top, SDS-PAGE of whole cell extracts showing the appearance of ~427 kDa dystrophin protein band (red arrowhead). Middle, Western blotting using anti-dystrophin mAb. Bottom, Western-blotting using control anti-tubulin mAb.

Supplementary Movie 1). Starting with an empty capsid shell purified from *E. coli* infected by the neck-minus and tail-minus T4 phage mutant (*10-amber.13-amber.HocΔ.SocΔ* T4) (Supplementary Fig. 1a), a penta-meric packaging motor was assembled on the portal vertex by simply adding the (monomeric) motor protein gp17 to the reaction mixture. The capsid interior is then filled with foreign DNA by adding linearized plasmid DNAs and ATP to the assembly reaction (Fig. 2a I, II). The T4 packaging motor captures DNA and translocates it into capsid from one end to the other in a processive fashion. This can repeat many times resulting in successive packaging of a series of DNA molecules

until the head is full (headful packaging)[51,60]. The packaging reactions were terminated by the addition of excess nuclease to digest the unpackaged DNA (Fig. 2a III, Supplementary Fig. 1b). Consequently, multiple copies of multiple plasmids are packaged inside the ~171 Kbp capacity T4 head (Supplementary Fig. 1b, c). Since the motor exhibits no sequence specificity, the composition of the packaged DNAs would be the same as that presented in the assembly reaction.

The packaged head particles would then be decorated with pro-teins, RNA, and their complexes through Soc and Hoc interactions (see below, Fig. 3a). Finally, and importantly, the particles are coated with

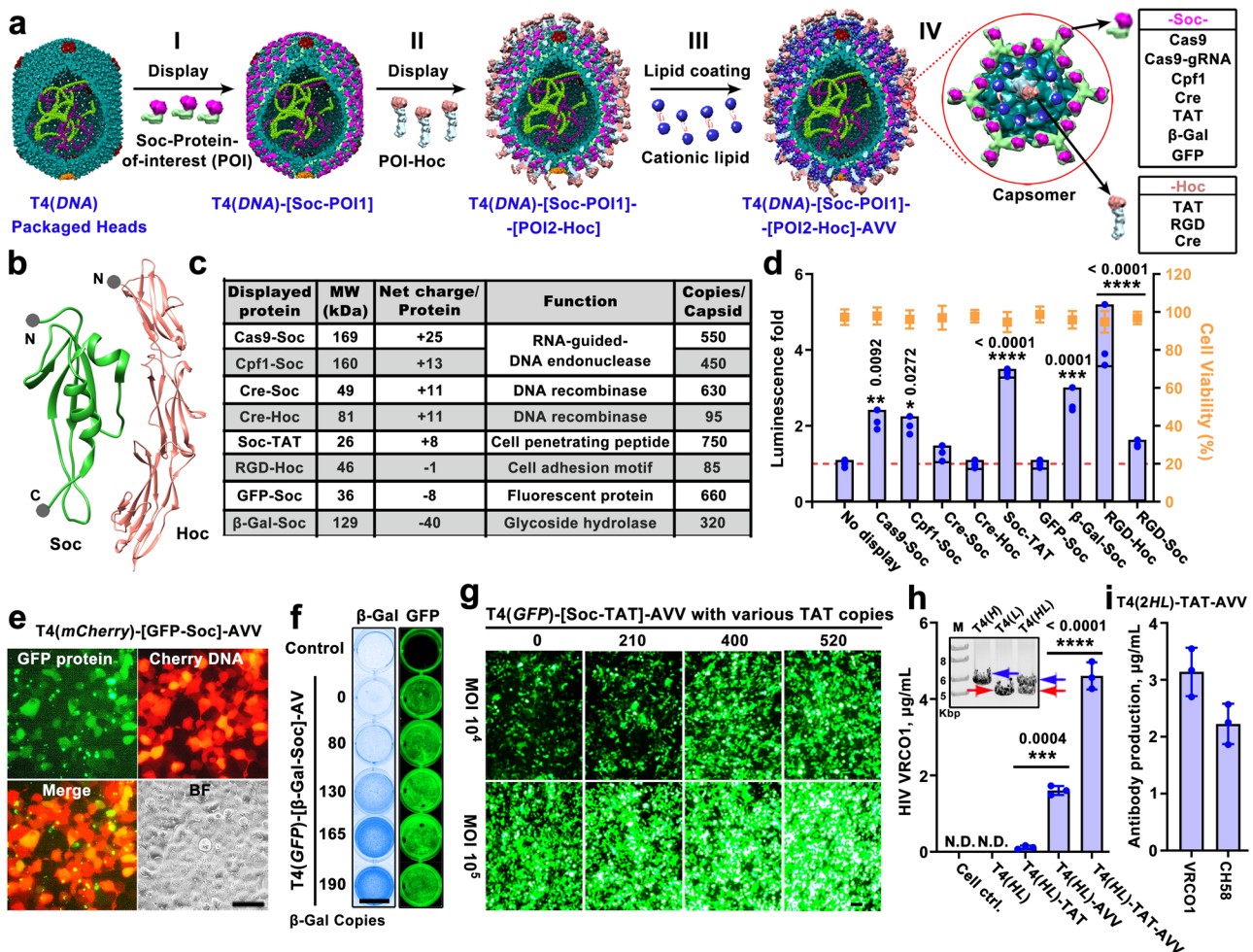

**Fig. 3 | Efficient co-delivery of genes and proteins by T4-AVVs. a** Schematic of sequential assembly to generate DNA-packaged and protein-displayed T4-AVVs. The DNA-packaged HocΔSocΔ heads as described in Fig. 2 were displayed with Soc- and/or Hoc- fused proteins and coated with cationic lipid (**I–IV**). POI, protein of interest. Enlarged capsomer (**IV**) shows various protein cargos carried by T4-AVVs via Soc and Hoc. **b** Structures of Hoc[34] (PDB 3SHS) and Soc[21] (PDB 5VF3) proteins showing the N and C termini used for fusing proteins of interest. **c** Table showing protein molecules with different sizes, net charges, and functions displayed on T4-AVVs and their copy numbers per capsid. **d** Comparison of T4(*Luci*)-AVV delivery using various displayed proteins, as determined by *Luci* expression (histogram) and cell viability (orange squares) (*n* = 3). T4(*Luci*)-AVV containing no displayed protein was used as control (dotted baseline). **e** Cells show fluorescence of internalized GFP protein (displayed) and transgene expression of *mCherry* DNA (packaged) following delivery by T4(*mCherry*)–[GFP-Soc]-AVVs. BF, bright field. Bar = 50 µm. **f** β-galactosidase (displayed protein) activity and *GFP* (packaged DNA) expression were examined following delivery by T4(*GFP*)-[β-Gal-Soc]-AVVs at increasing copy numbers of displayed β-galactosidase-Soc. Bar = 15 mm. **g** Fluorescence images depict enhanced delivery and *GFP* transgene expression of TAT-displayed T4(*GFP*)-AVVs with increasing copy number of TAT at a MOI of 10⁴ or 10⁵. Bar = 50 µm. **h** HIV gp120 envelope protein-specific ELISAs to quantify the amount of VRCO1 broadly neutralizing antibody (bnAb) secreted by 293T cells 48 h following transduction (*n* = 3): medium buffer (Cell ctrl.); T4(*HL*) (naked, no TAT), T4(*HL*)-TAT (Soc-TAT decorated), T4(*HL*)-AVV (cationic lipid coated), and T4(*HL*)-TAT-AVV (Soc-TAT and lipid coated). The inset shows the packaging of *VRCO1* expression plasmids encoding heavy chain H (blue arrow), light chain L (red arrow), and H + L chains. **i** Quantification of secreted VRCO1 and CH58 bnAbs by 293T cells following transduction with T4(*VRCO1-HL* + *CH58-HL*)-[Soc-TAT]-AVVs (Soc-TAT and lipid coated) (*n* = 3). Values represent mean with SD. N.D., not detected, *\*P* < 0.05, *\*\*P* < 0.01, *\*\*\*P* < 0.001, and *\*\*\*\*P* < 0.0001, one-way ANOVA test.

cationic lipid molecules (Fig. 2a IV). Since the T4 capsid has a high density of negative charges, ~6,829 net negative charges per capsid (Fig. 1f), we hypothesized that cationic lipids would spontaneously assemble on T4 capsid via electrostatic interactions. Indeed, extensive lipid binding occurred when cationic lipids were added to T4 capsids and negative-stain electron microscopy (EM) showed a diffused stain around the capsid (Supplementary Fig. 2a–c). Additionally, no evidence of leakage of packaged DNA was observed by EM (Supplementary Figs. 1b, 2a). When labeled with fluorophores, these enveloped particles showed co-localization of the capsid-labeled Alexa fluorophore and the lipid-labeled NBD fluorophore (Fig. 2b, Supplementary Fig. 2b, c). The T4-AVV nanoparticles thus assembled possess the basic architecture of naturally enveloped viruses with lipid coat, capsid shell, and packaged "genome" (Supplementary Movie 1).

## The T4-AVVs efficiently deliver genetic payloads into human cells

We reasoned that the T4-AVVs by virtue of their positively charged lipid coat should bind to the negatively charged and lipophilic surface of human cells and allow entry. Several cationic lipids and cell penetration peptides have been well-documented to exhibit such a property[61,62]. Indeed, a series of experiments demonstrated that the lipid-coated T4-AVVs efficiently delivered genetic payloads into human cells. For instance, when co-packaged with two different linear plasmids, on average ~5 molecules each of *GFP* reporter plasmid (5.4 Kbp) and *luciferase* plasmid (*Luci*, 6.3 Kbp) per nanoparticle (Supplementary Fig. 1c), these AVVs were able to transduce the reporter plasmids into HEK293T (293T) cells with near 100% efficiency, as determined by the percentage of cells expressing GFP fluorescence (Fig. 2c, d).

Under optimal conditions (Supplementary Fig. 2d, e), the luciferase activity of T4(*Luci*)-AVVs was ~$10^5$-fold higher than the naked capsids lacking the cationic lipid coat (T4(*Luci*)), and ~$10^2$-fold higher than the capsids that are cationic but lacked the lipid (T4(*Luci*)-TAT) (Fig. 2d). The latter capsids were generated by displaying a cationic cell penetration peptide, HIV-TAT (NGYGRKKRRQRRRG)[63]. Relatively low amounts of lipids were sufficient to coat the capsids and no significant cell toxicity was observed at the maximum signal read-out of delivered transgene expression (Supplementary Fig. 2f, g). No major differences were observed among various cationic lipids tested, although LPF2K and LPFRNAiMAX gave the highest transduction efficiencies (Fig. 2d). In contrast, a T4 (*Luci*) control (no lipid or TAT), or the T4-AVV unpackaged control (no ATP in the reaction), or the cells-only buffer control, produced no significant signal (Fig. 2d). These controls also demonstrated that the signal was due to the T4-AVVs but not due to any nonspecific lipid encapsulated, undigested, or leaked DNA.

A head-to-head comparison of the T4-AVV transduction with that of the most widely used AAV2[12], both packaged with the same-sequence plasmid (*AAV2ITR-CMV enhancer and promoter-fireflyLuci-hGH polyA*) (Supplementary Fig. 2h), showed 4-19 fold greater expression of *luciferase* by T4-AVVs when compared to AAV2 (Fig. 2e). This could be due to T4's ability to package ~8 molecules of *Luci* plasmid in each head, delivering multiple copies of the genetic payload in a single transduction, while AAV2 is limited to one copy at a time. The difference was greatest at low MOI (ratio of capsid particles to cells), which decreased with increasing MOI (Fig. 2e). This is because, at high MOI ($10^5$), the cells may have reached the maximum transduction capacity for the T4-AVVs, whereas transduction by AAV continues to increase proportionally thereby reducing the fold-difference in the ratio of the luciferase activity between T4-AAV and AAV transductions (Supplementary Fig. 2f).

A key distinguishing feature of T4-AVVs is its large capacity head which can accommodate up to ~171 Kbp, hence it can deliver large therapeutic genes such as the ~11 Kbp human *dystrophin* gene (full-length coding DNA), which none of the current AAV and LV vectors are able to. In fact, the *dystrophin* gene is one of the largest known human genes[64] and mutations in dystrophin lead to Duchenne muscular dystrophy (DMD), a debilitating degenerative muscular disease that causes early death. To demonstrate the applicability of T4-AVVs for large gene delivery, we selected a ~20 Kbp *dystrophin-reporter* plasmid consisting of ~17 Kbp *dystrophin* gene cassette under the control of the CMV promoter. The cassette included, in addition to the full-length 11 Kbp *dystrophin* gene (*Dys*), three reporter genes encoding *luciferase*, *puromycin N-acetyl-transferase (Pac*, or puromycin resistant gene*)*, and *mCherry* tandemly fused inframe via a cleavable P2A peptide[65] (Fig. 2f). The T4-AVVs packaged the *Dys* plasmid (~2.5 molecules per capsid on average; Fig. 2g and Supplementary Fig. 1c) and delivered it into 293T cells (Fig. 2h, i). Consequently, the cells simultaneously became mCherry-positive, luciferase-positive, and puromycin-resistant (Fig. 2h). SDS-PAGE showed the appearance of a strong 427 kDa dystrophin protein band in the transduced cells in a dose-dependent manner but not in the controls. That this band was dystrophin was ascertained by Western blotting (Fig. 2i) and mass spectrometry (Supplementary Table 1). Finally, the utility of T4-AVVs for large DNA packaging was further demonstrated by packaging the 40 Kbp T7 phage genomic DNA (~2 molecules per head; Supplementary Fig. 1c).

## Co-delivery of genes and proteins by T4-AVVs

To incorporate protein payloads into T4-AVVs, Soc- and/or Hoc-fused proteins were added to the packaging reaction mixture (Fig. 3a I, II). By virtue of the symmetrically positioned Soc and Hoc binding sites, these molecules formed arrayed assemblies on capsid lattice (Fig. 3a IV). At ~10-20:1 ratio of molecules to binding sites, full occupancy, i.e., up to 870 Soc- and 155 Hoc-fused proteins per capsid, was observed (Supplementary Fig. 3a to 3c). The unbound proteins were then removed by

centrifugation and the pelleted T4 head nanoparticles were complexed with lipids by fast mixing to complete the AVV assembly (Fig. 3a III, IV).

To determine if these T4-AVVs can now co-deliver proteins along with genes, a series of AVVs were assembled by displaying proteins fused to either the N- or C-terminus of Soc or the N-terminus of Hoc (Fig. 3b). These termini are solvent-exposed in our solved atomic structures[21,22,33,34], hence are highly suitable for display, while the C-terminus of Hoc due to its interaction with the capsid is less suitable. A series of proteins with different sizes, charges, oligomeric states, and functions were incorporated into T4-AVVs, as follows: 320 molecules of β-Gal-Soc (129 kDa; −40 charges) (net charges per molecule), 660 molecules of GFP-Soc (36 kDa; −8 charges), 85 molecules of RGD-Hoc (46 kDa; −1 charge), 710 molecules of Soc-RGD (10 kDa; −1 charge), 750 molecules of Soc-TAT (26 kDa; +8 charges), 630 molecules of Cre-Soc (49 kDa; +11 charges), 95 molecules of Cre-Hoc (81 kDa; +11 charges), 450 molecules of Cpf1-Soc (160 kDa; +13 charges), and 550 molecules of Cas9-Soc (169 kDa; +25 charges) (Fig. 3c, Supplementary Fig. 3a–c, and Supplementary Note 1). To ensure accurate quantification and appropriate head-to-head comparisons, the same batch of purified T4 heads (~$10^{14}$ T4 heads from 1 L *E. coli* culture) was used for bulk DNA packaging (*Luci* plasmid) in a single tube and the packaged heads were then equally divided to display various proteins and their transduction efficiencies at same MOIs were determined.

Somewhat unexpectedly, protein-displayed AVVs, in general, showed enhanced transduction efficiency (up to 4-fold) when compared to AVVs lacking a displayed protein, as measured by the delivered luciferase activity (Fig. 3d). Luciferase was used as an internal control in these experiments because the same AVVs containing the same number of head particles and packaged *Luci* plasmid DNA molecules were used for testing each variable. The enhanced transduction was probably because the displayed protein molecules contributed additional charges that resulted in better lipid coating and/or cell binding.

Additionally, the 9-aa disulfide-constrained RGD peptide (CDC**RGD**CFC), a cell surface targeting ligand, when fused to the tip of Hoc fiber also showed delivery enhancement (Fig. 3d). This tripeptide motif has been well-documented to bind to the abundantly present integrin molecules on human cells[66]. Furthermore, the luciferase activity of such RGD-Hoc AVVs was ~2.5 to 4-fold higher than the RGD-Soc AVVs (Fig. 3d), even though the copy number of Soc-RGD was ~8-fold greater than that of RGD-Hoc. It appears, therefore, that the targeting ligand attached to the tip (N-terminus) of ~185Å-long flexible Hoc fiber imparted much greater reach to capture the integrin receptor molecules for the lipid-coated T4-AVVs than the Soc-fused RGD that is closely associated with the capsid wall.

Remarkably, the T4-AVVs co-delivered the displayed proteins as well as the packaged plasmids in a functional state. For instance, when exposed to 293T cells, the GFP-displayed AVVs showed strong green fluorescence, initially at the cell surface (~3 h) and then throughout the cell (~20 h). When the same AVVs were also packaged with *mCherry* reporter plasmid, the cells in addition began showing red fluorescence at 6 h and continued to intensify up to 48 h, due to the expression of the delivered *mCherry* gene (Fig. 3e, Supplementary Fig. 3d, e). Similarly, cells transduced with AVVs displaying ~516 kDa tetrameric β-galactosidase (β-Gal) and packaged with *GFP* or *Luci* reporter plasmids, exhibited β-galactosidase activity in a dose-dependent manner (Fig. 3f), as well as the luciferase activity and GFP fluorescence (Fig. 3d, f).

Notably, positively charged proteins such as TAT, Cas9, and Cpf1 showed greater enhancement, with the TAT-AVVs displaying 520 copies of TAT showing the greatest, ~3.5-fold, enhancement in a dose-dependent manner (Fig. 3d, g). In addition to reporter genes, we have also assembled TAT-AVVs packaged with two therapeutically relevant expression plasmids (Supplementary Fig. 1c) carrying heavy (H) and

light (L) chain genes of VRC01 IgG antibody, a potent broadly neutralizing antibody (bnAb) against HIV-1[67]. An average of 10-12 molecules of H and L plasmids were packaged per capsid (Supplementary Fig. 1c and Fig. 3h). These T4(*VRC01 HL*)-[Soc-TAT]-AVVs co-transduced and co-expressed the H and L chains, as evident from the secretion of functional IgG molecules at high levels (~4.5 mg/liter). The bnAb levels were 3-fold higher than the T4(*HL*)-AVVs lacking TAT, and 20-fold higher than the T4(*HL*)-[Soc-TAT] lacking the lipid coat (Fig. 2h). Naked particles T4(*HL*) having neither TAT nor lipid coat produced undetectable levels of the antibody. Furthermore, we assembled AVVs that delivered two H and two L chains belonging to two different HIV-1 bnAbs, VRC01 and CH58, into the same capsid. Here, an average of ~11 molecules, mixtures of four different plasmids, were packaged into the same capsid (Supplementary Fig. 1c). Such TAT-AVVs co-transduced these plasmids into 293T cells and secreted both VRC01 (~3 mg/liter) and CH58 (~2 mg/liter) bnAbs into the culture medium (Fig. 2i).

Together, the above sets of data demonstrated that co-delivery of (multiple) genes and proteins is another key distinguishing feature of the T4-AVVs, which can be exploited to program AVVs by custom engineering of payloads for various applications.

## Genome editing AVVs

The ability to program AVVs with combinations of genes and proteins can be used to perform complex molecular operations in human cells, which would open a vast array of therapeutic applications[15,68–70]. With this in mind, we assembled a variety of genome editing AVVs by incorporating all the editing molecules into one AVV in different configurations (Fig. 4a, b). Cryo-EM showed the presence of genome editing complexes decorating the capsid (Fig. 4c).

First, we assembled AVVs packaged with plasmids carrying expressible Cas9 and gRNA genes under the control of the CMV and U6 promoters, respectively. The Cas9 sequence was codon-optimized and fused with the nuclear localization sequence (NLS) PKKKRKV[71] at its N-terminus (NLS-Cas9). This allowed transport of Cas9 into the nucleus to carry out genome editing. The gRNA was targeted to the PPP1R12C locus on chromosome 19 of the human genome, also known as the AAVS1 safe harbor locus[72]. On average, each capsid was packaged with 7 molecules of the 8.3 Kbp plasmid containing both these expression cassettes (AVV1). In a second configuration (AVV2), we assembled AVVs by incorporating purified Cas9 as displayed protein fused to Soc (NLS-Cas9-Soc, 169 kDa) (Supplementary Fig. 4a, b), while the gRNA was supplied as a packaged plasmid. Up to ~550 molecules of NLS-Cas9-Soc could be displayed on the surface (Fig. 4d, top) and ~10 copies of gRNA plasmid were packaged inside the capsid (Supplementary Fig. 1c). Additionally, a series of biochemical assays were performed to ensure that the NLS-Cas9-Soc and gRNAs exhibited full functionality, i.e., formation of ribonucleoprotein (RNP) complexes and gRNA-directed cleavage of target DNA (Supplementary Fig. 4c, d). We have also packaged a second GFP reporter plasmid into both these AVVs that served as a marker for transduction efficiency. It also confirmed that the display of NLS-Cas9-Soc, a cationic protein, markedly enhanced the AVVs' delivery efficiency (Fig. 4e), in a dose-dependent manner (Supplementary Fig. 4e). These AVVs when transduced into 293T cells carried out genome editing, also in a Cas9 dose-dependent manner, by introducing double-stranded breaks at the targeted AAVS1 locus followed by repair by non-homologous end joining (NHEJ) which created short insertions and deletions (indels) at the target site, as determined by T7 Endonuclease I assay (T7EI) and DNA sequencing (Supplementary Fig. 4f). The efficiency of genome editing by AVV1 and AVV2 was ~12 to 15% (Fig. 4f, Supplementary Fig. 4f).

We then assembled AVVs by incorporating Cas9 and gRNA as a pre-formed RNP complex (AVV3). About 280 copies of ~210 kDa Cas9-gRNA RNP complex were displayed on the capsid through Soc (Fig. 4c,

d, Supplementary Fig. 4g). In the fourth configuration, we have in addition packaged ~7 molecules of Cas9-gRNA expression plasmid (AVV4). In both cases, either *GFP* or *Luci* reporter plasmids were also packaged to confirm high efficiency of transduction (Fig. 4e and Supplementary Fig. 4h). These, AVVs 3 and 4, gave the best editing efficiencies, ~30-35% indels at the AAVS1 locus, about twice that obtained by lipofectamine transfection, which was used as an external positive control (Fig. 4f and Supplementary Fig. 4i).

To perform genome editing at a therapeutically important site, we targeted the T4-AVVs to the hemoglobin beta gene (HBB) on chromosome 11 of the human genome. These AVVs assembled with Cas9-HBB gRNA RNP complexes (AVV4) performed ~20-25% editing at this site (Supplementary Fig. 4j). Furthermore, we have assembled AVVs by displaying two gRNA-RNP complexes, one targeted to HBB site and the other to AAVS1 site, on the same AVV (Fig. 4g) (AVV5). These AVVs were also packaged with ~7 molecules of Cas9 and HBB/AAVS1gRNA expression plasmids. These multiplex AVVs successfully performed genome editing of both the targeted sites present on two different chromosomes, ~20% at the HBB site and ~30% at the AAVS1 site (Fig. 4g).

## Gene recombination AVVs

Next, we asked whether the T4-AVVs can perform genome editing as well as gene recombination, homologous or site-specific, in the same cell. Previously it was reported that Cas9-generated DNA breaks facilitate homologous recombination near the cleaved site[73,74]. AVVs were assembled by displaying AAVS1-targeted gRNA-NLS-Cas9-Soc RNP complexes on capsid and packaging a donor plasmid containing either a 6.6 Kbp promoter-less *Pac* (puromycin resistant gene) (Donor 1), or a much larger 11.1 Kbp promoter-less *Pac* followed by CMV promoter-driven SARS-CoV-2 spike ectodomain gene (*Pac-CMV-S-ecto*, Donor 2). Each donor plasmid was also engineered by adding ~800 bp homologous arms flanking the Cas9 cleavage site to either end of the transgenes (Fig. 4h and Supplementary Fig. 5a). Puromycin resistance will emerge if homologous recombination occurred following Cas9 cleavage, bringing the *Pac* or *Pac-CMV-S-ecto* genes under the control of an upstream AAVS1 promoter (Fig. 4h). Indeed, puromycin resistance clones arose following transduction by these AVVs, whereas control AVVs lacking the RNP complex showed no puromycin resistance. PCR and DNA sequencing showed that 15 out of 15 single-cell clones exhibiting puromycin resistance contained *Pac* gene insertion precisely at the Cas9 cleavage site (Fig. 4i and Supplementary Fig. 5b). Furthermore, in the case of *Pac-CMV-S-ecto* recombination, stable cell lines with *Pac-CMV-S-ecto* knock-in at the targeted site were generated, which secreted high levels of S-ecto trimer, the antigen component of the COVID-19 vaccines (Fig. 4j). The functionality of S-ecto trimer was ascertained by its binding to human ACE2 receptor (Fig. 4k), which demonstrated comparable activity to that produced by transient transfections[38,40].

We then assembled AVVs programmed with an even more complex set of payload molecules to demonstrate site-specific recombination. The capsids were displayed with 95 copies of the site-specific recombinase Cre as Hoc fusion protein (Fig. 4l, Supplementary Figs. 3b and 5c–e), and the *Luci* reporter plasmid was packaged inside the capsid as an internal control for transduction efficiency. Additionally, these capsids also carried the gRNA-Cas9-Soc RNP complexes displayed on the surface (Fig. 4l, Supplementary Fig. 5f, g). These AVVs programmed with four different payload molecules; Cre recombinase, Cas9 nuclease, gRNA, and *Luci* reporter plasmid, were then transduced into a 293 T cell line containing the *LoxP-mCherry-stop-LoxP* cassette upstream of a promoterless *GFP* reporter gene (Fig. 4l). We observed several genome modifications in these cells. First, Cre-mediated site-specific recombination occurred, as evident by the cells showing strong red fluorescence but no green fluorescence at the start due to endogenous mCherry expression, which then turned into intensely

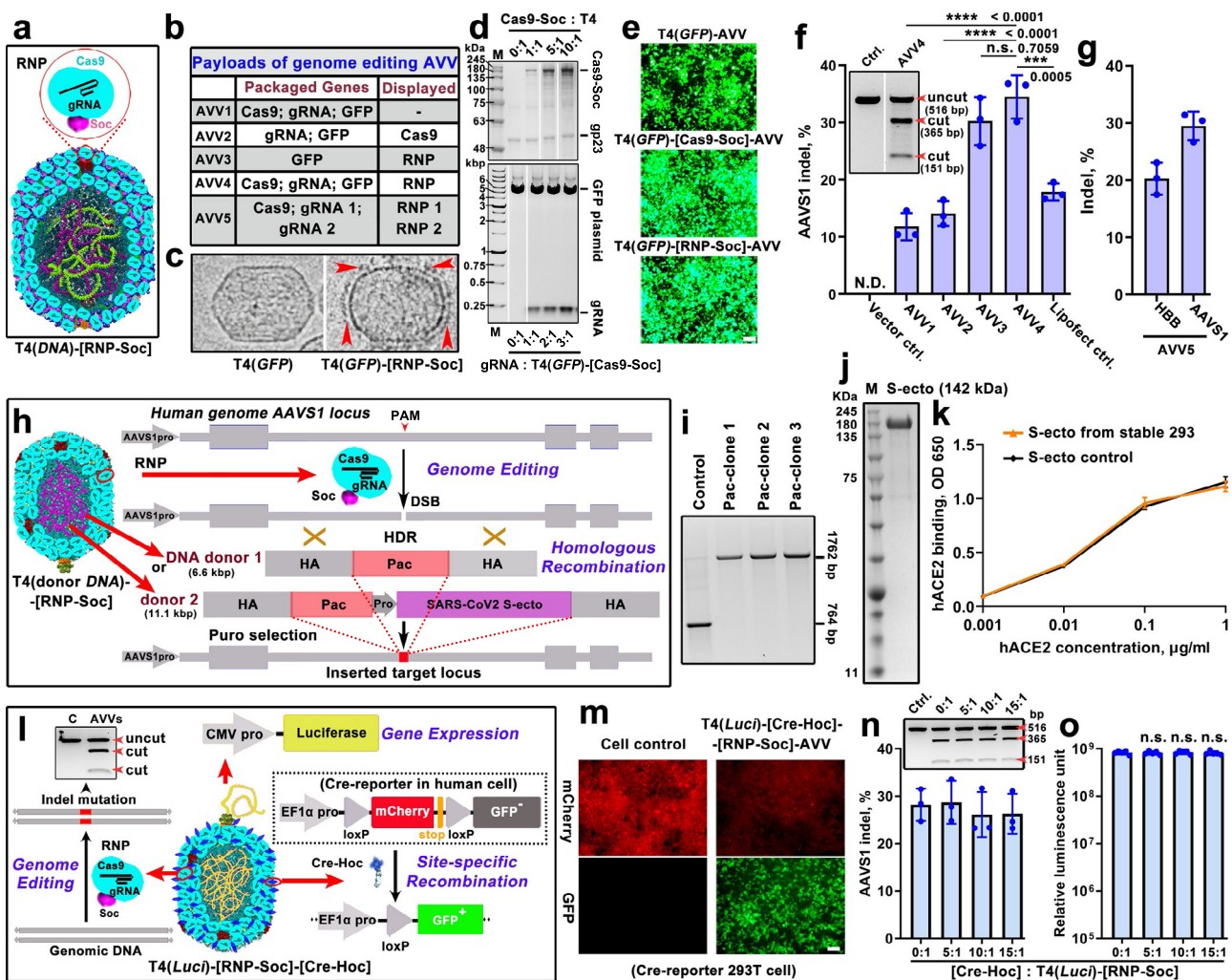

**Fig. 4 | Genome editing and recombination T4-AVVs. a** Schematic illustration of genome editing AVVs. **b** Compositions of payload molecules in genome editing AVVs 1-5. Ribonucleoprotein (RNP) consists of NLS-Cas9-Soc protein and guide RNA (gRNA). **c** Cryo-EM images of control T4(*GFP*) and genome editing T4(*GFP*)-[RNP-Soc] heads. The red arrowheads indicate the displayed RNP. **d** Top: SDS-PAGE showing the display of NLS-Cas9-Soc on T4 capsid at increasing ratios of NLS-Cas9-Soc molecules to Soc-binding sites. Bottom: Agarose gel electrophoresis showing increasing binding of gRNA to T4(*GFP*)-[Cas9-Soc] capsids with increasing ratios of gRNA molecules to Soc binding sites. **e** Fluorescence microscopy images showing the comparison of *GFP* expression in 293 T cells transduced with T4(*GFP*)-AVVs, T4(*GFP*)-[Cas9-Soc]-AVVs, and T4(*GFP*)-[RNP-Soc]-AVVs. Bar = 50 μm. **f** Comparison of genome editing efficiencies as determined by % AAVS1 indels generated by delivery of T4-AVVs 1–4 (*n* = 3). The inset shows the representative T7E1 assay of AAVS1 indel frequencies of cells treated with AVV4. **g** Simultaneous genome editing at two distinct target sites on the human genome by T4-AVV5 carrying two gRNAs as RNP in the same AVV (*n* = 3). **h** T4-AVVs carrying out genome editing and homologous recombination at the AAVS1 locus by delivering RNP (displayed) and donor *Pac* or *Pac-S-ecto* (SARS-CoV-2 spike ectodomain) plasmid DNA (packaged). **i** PCR assay on puromycin-resistant single cell clones following transduction with T4(*Pac*-donor)-[RNP-Soc]-AVVs. **j** SDS-PAGE showing the 142 kDa SARS-CoV-2 S-ecto protein purified from the stable Pac-S-ecto 293T cells which were produced by transduction with T4(*Pac-S-ecto* donor)-[RNP-Soc]-AVVs and screened by puromycin. **k** ELISA-based functional analysis of S-ecto trimer (from stable 293T cells) binding to human ACE2 receptor (*n* = 3). The positive control S-ecto trimer was purchased from Sino Biological. **l** Schematic of site-specific recombination by T4(*Luci*)-[RNP-Soc]-[Cre-Hoc]-AVVs. **m** AVVs mediated site-specific recombination in Cre reporter cells. Bar = 50 μm. **n, o** T7E1 assay and quantification of AAVS1 indel frequencies (**n**) (*n* = 3) and luciferase activity (**o**) (*n* = 5) of 293T cells treated with T4(*Luci*)-[RNP-Soc]-[Cre-Hoc]-AVVs with increasing amounts of displayed Cre-Hoc. Values represent mean with SD. ***P < 0.001 and ****P < 0.0001, n.s., not significant, one-way ANOVA test.

green fluorescence while the red fluorescence faded (Fig. 4m). This means that the transcriptionally active *mCherry* gene was spliced out by intramolecular recombination between the flanking LoxP sites by the AVV-delivered Cre, which activated the GFP reporter expression that now came under the control of an upstream promoter due to the site-specific recombination (Fig. 4l, m). Second, these AVVs also carried out genome editing at a different site by the delivered Cas9 and gRNA, as evident from ~30% gene disruption at the AAVS1 locus (Fig. 4n). Third, the AVVs co-delivered the *Luci* reporter gene resulting in high-level expression of the transgene. Furthermore, our analyses show that Cre-Hoc display or its copy number did not interfere with the delivery of this complex payload (Fig. 4o).

Finally, the high efficiency of site-specific recombination by T4-AVVs was also verified by another approach (Supplementary Fig. 5h). We constructed AVVs with another large payload: 95 molecules of displayed Cre-Hoc, 270 molecules of gRNA-Cas9-Soc RNP complex, 5 molecules of packaged *mCherry* reporter plasmid, and 6 molecules of packaged *GFP* donor plasmid. The latter contained the *CMV promoter-LoxP-polyA STOP-LoxP* cassette upstream of the *GFP* reporter gene. Remarkably, these AVVs efficiently performed all the tasks they were programmed with (Supplementary Fig. 5h, i). The Cre protein performed LoxP recombination on the co-delivered *GFP* donor plasmid, splicing out the polyA transcriptional STOP sequence and bringing the *GFP* reporter under the control of the upstream CMV promoter

producing green fluorescence, in a Cre-dose dependent manner (Supplementary Fig. 5j). In addition, virtually every cell showed *mCherry* expression in addition to green fluorescence (Supplementary Fig. 5i) and ~30% genome editing at the AAVS1 site (Supplementary Fig. 5h).

## RNA-delivering AVVs

In light of the strong interaction observed between Cas9 and gRNA and efficient delivery of the resultant RNP complexes by T4-AVVs (Fig. 4 and Supplementary Fig. 4), we asked if this system could be adapted for RNA delivery by including siRNAs, potentially hundreds of copies of multiple siRNAs, as part of the same payload (Fig. 5a). siRNAs are

~20–25 bp double-stranded oligonucleotides that target mRNA(s) having the same sequence for degradation instead of translation. Such siRNA-mediated gene silencing mechanism has been extensively used for the treatment of various genetic and infectious diseases[75].

Our results show that Cas9 bound to siRNAs similar to gRNA (Supplementary Fig. 6a). Consequently, the T4-AVVs could be decorated with ~280 Cas9-siRNA RNP complexes (Fig. 5a, b). When exposed to 293T cells, these AVVs which also contained the packaged *GFP* or *Luci* reporter plasmids (Fig. 5c and Supplementary Fig. 6b) delivered siRNA molecules and silenced GFP expression, in a dose-dependent manner, while the control AVVs delivering a nonspecific control siRNA (NCsiRNA) had no effect (Fig. 5c, 5d, and Supplementary Fig. 6c).

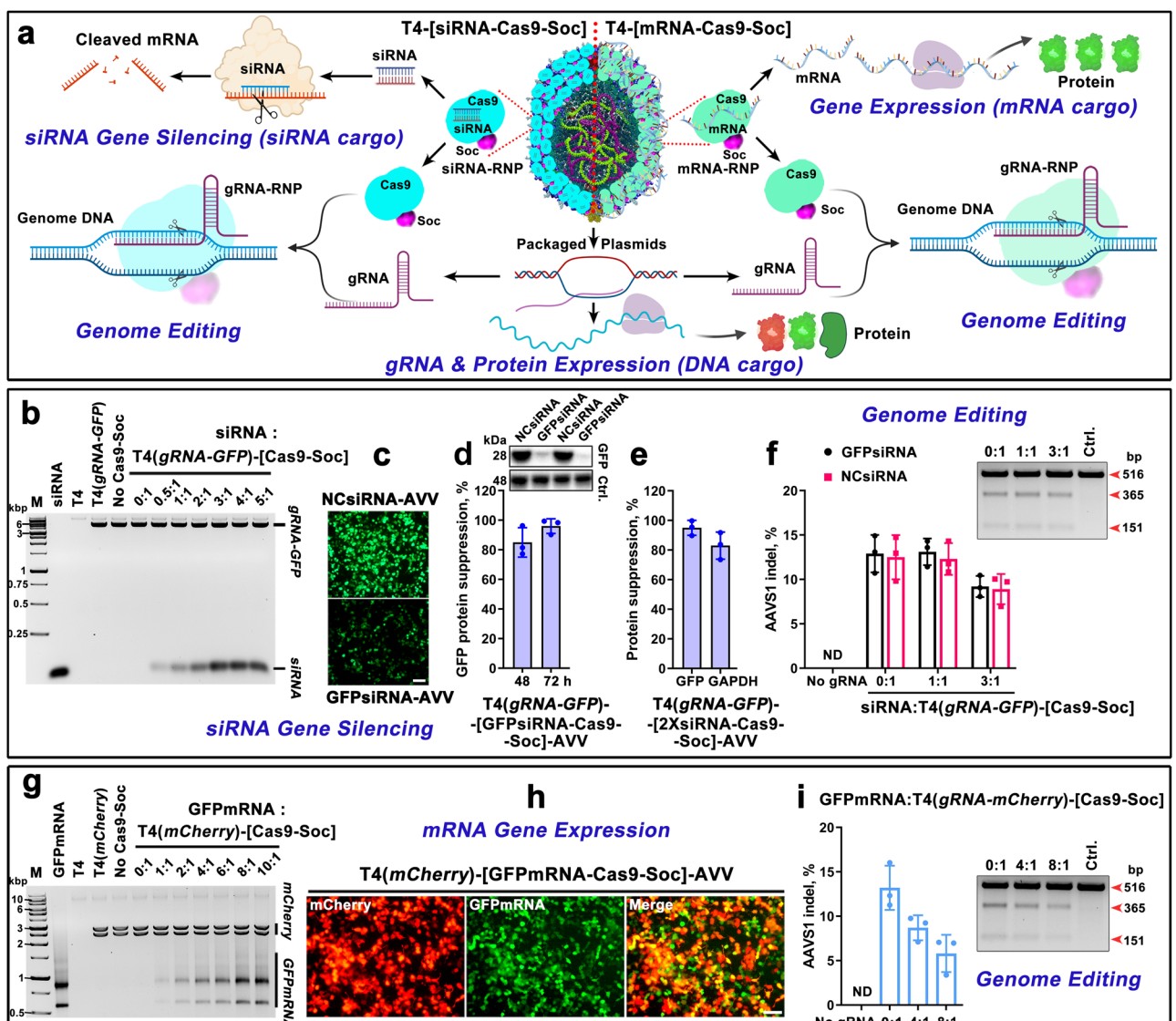

**Fig. 5 | RNA delivering T4-AVVs. a** Schematic illustration of T4-AVVs carrying siRNA and mRNA on the capsid surface as complexes of Cas9-Soc. These AVVs can carry out multiple genetic operations upon delivery; gene silencing (siRNA cargo), genome editing (RNP cargo), and gene expression (DNA and mRNA cargo). **b** Agarose gel showing the binding of siRNA to T4(*gRNA-GFP*)-[Cas9-Soc] capsids at increasing ratios of siRNA molecules to Soc binding sites. The positions of the packaged *gRNA-GFP* plasmid and the displayed siRNA are indicated. **c** Silencing (knock-down) of GFP expression in 293T cells treated with T4(*gRNA-GFP*)-[GFPsiRNA-Cas9-Soc]-AVVs (GFPsiRNA-AVVs). The control T4(*gRNA-GFP*)-[NCsiRNA-Cas9-Soc]-AVVs (NCsiRNA-AVVs) carry a nonspecific siRNA. Bar = 50 μm. **d** Western blotting quantification using a GFP-specific mAb showing suppression of GFP protein levels by GFPsiRNA-AVVs at 48 and 72 h post-transduction (*n* = 3).

**e** Simultaneous silencing of two different genes by incorporating two siRNAs into the same AVV, one targeted to *GFP* gene and the other to endogenous *GAPDH* gene (*n* = 3). **f** AAVS1 indel frequencies of cells treated with T4(*U6pro-AAVS1gRNA-CMVpro-GFP*)-[siRNA-Cas9-Soc]-AVVs at increasing siRNA ratios (*n* = 3). The inset at the top right shows AAVS1 gene disruption using T7E1 assay. **g** Agarose gel showing the binding of GFPmRNA to T4(*mCherry*)-[Cas9-Soc] capsids at increasing ratios of mRNA molecules to Soc binding sites. The positions of the packaged *mCherry* plasmid and the displayed GFPmRNA bands are indicated. **h** Dual expression of AVV-packaged *mCherry* plasmid DNA and AVV-displayed GFPmRNA in the same cell. Bar = 50 μm. **i** Genome editing at AAVS1 locus by T4(*U6pro-AAVS1gRNA-CMVpro-mCherry*)-[GFPmRNA-Cas9-Soc]-AVVs at increasing mRNA ratios (*n* = 3). Values represent mean with SD.

Remarkably, up to ~90% silencing was achieved in 48 h, and nearly 100% silencing in 72 h, as assessed by Western blotting using a GFP monoclonal antibody (Fig. 5d). Furthermore, two siRNAs silencing different mRNAs could be simultaneously delivered. Accordingly, T4-AVVs carrying GFP-siRNA and GAPDH-siRNA knocked down the expression of both these genes, by ~95% and 80%, respectively (Fig. 5e).

Delivery of much longer mRNA molecules would further expand the footprint of RNA-AVVs to high-level expression of genes for therapeutic applications[76,77]. To test this, the siRNA in T4-AVVs was replaced with mRNA by mixing the in vitro transcribed 996-nt GFP mRNA with T4-[Cas9-Soc] capsids. The Cas9-mRNA complexes formed on the capsid, reaching saturation at ~8:1 ratio of mRNA molecules to AVV-displayed Cas9 (Fig. 5a, g, and Supplementary Fig. 6d). Each AVV carried a payload of ~55 molecules of mRNA. The lower copy number of mRNA, when compared to siRNA, was probably because the much longer mRNA titrated several molecules of Cas9. No significant mRNA binding was evident with different versions of the control T4 capsids lacking Cas9 (Fig. 5g). These GFPmRNA-AVVs upon transduction into 293T cells expressed strong green fluorescence in the cells (Fig. 5h). The fluorescence was evenly distributed throughout the cell and merged with the red fluorescence generated by co-delivery and expression of *mCherry* reporter gene packaged as plasmid in the same AVV. On the other hand, control (mCherry)AVVs lacking the Cas9-mRNA complex showed only red fluorescence (Supplementary Fig. 6e). Additionally, comparable expression of packaged *Luci* reporter among AVVs displaying various amounts of siRNA or mRNA suggested that RNA display did not significantly affect the efficiency of AVV transduction (Supplementary Fig. 6b, f).

To further enhance the utility of the RNA-AVVs, we packaged another gRNA expression plasmid into the above AVVs (Fig. 5a). The idea was that, upon delivery, the displayed Cas9 can first deliver siRNA or mRNA into the cytosol and then, by virtue of the fused NLS sequence, it can relocate to the nucleus and form a genome editing complex with the gRNA transcribed from the co-delivered plasmid. The Cas9-gRNA complex can then perform a second function, genome editing at a target site (Fig. 5a). Control in vitro experiments showed that gRNA can replace bound siRNA in the Cas9-siRNA complex, whereas the reverse, siRNA replacement of gRNA in the Cas9-gRNA complex, was inefficient (Supplementary Fig. 6g, h). We took advantage of these differential affinities of Cas9 to siRNA and gRNA and programmed the AVVs with five different components (Fig. 5a): displayed Cas9-siRNA or Cas9-mRNA complexes and packaged gRNA and *GFP/mCherry* reporter plasmids. Upon transduction into 293 T cells, these AVVs performed all the functions they are programmed with; *GFP* gene silencing (siRNA), *GFP* gene expression (mRNA), *mCherry* expression (plasmid), and genome editing (Cas9 and gRNA) at AAVS1 in the same cell (Fig. 5f, i). However, the editing efficiency decreased with increasing amounts of bound siRNA and mRNA, reflecting reduced availability of free Cas9 (Fig. 5f, i),

## Re-wiring the capsid exterior

Next, we asked if the T4 capsid can be re-wired by CRISPR engineering such that it can be further programmed with a large number of foreign peptides, 930 or one per gp23* capsid protein subunit, on the capsid exterior. Such a functionalized surface could endow T4-AVVs with properties to overcome intracellular barriers for payload delivery; endosome escape, nuclear localization, etc. Preliminary studies using various inhibitors showed that the T4-AVVs are internalized through dynamin- and clathrin-mediated endocytosis, and microtubule stabilizing agents such as Tubastatin A (TBA) facilitate the transport of delivered DNA from cytosol to nucleus.

Our cryo-EM structures of T4 head[21,22] showed a flexible, solvent-exposed, external loop in the insertion domain (I domain) of the major capsid protein gp23* (Supplementary Fig. 7a, b). We hypothesized that AVV nanoparticles exposing additional acidic peptides inserted into

this loop might be able to alter the pH of endosomes and help disrupt the endosomal membrane for escape, as was observed by the pH-sensitive membrane-perturbing peptides such as GALA and INF7[78–80] and those found in human viral pathogens such as influenza virus and flock house virus[81,82]. Furthermore, acidified capsid surface would allow more efficient cationic lipid coating of the T4-AVVs.

Peptides of various lengths consisting of aspartic acid (D) and glutamic acid (E) residues were inserted into the I domain loop by CRISPR engineering (Fig. 6a and Supplementary Fig. 7a–c). Remarkably, the T4 capsid can tolerate up to 9 acidic residues (9DE) with no significant effect on plaque size or phage production. We then introduced *9DE-gp23*, *10am*, *13am*, *Hoc-del*, and *Soc-del* mutations into the genome through CRISPR to generate 9DE-T4 phage (*9DE-gp23.10-amber.13-amber.HocΔ.SocΔ* T4) and heads for AVV assembly. A 3.9 Å high-resolution cryo-EM structure of 9DE-T4 head was generated (Fig. 6b, c, Supplementary Table 2, and Supplementary Note 2), which showed decoration of the capsid with 9DE peptides between residues G201 and A202 of the I domain loops (Fig. 6d). The six 9DE peptides in each capsomer formed a negatively-charged acidic ring (Fig. 6b), generating an extraordinarily charged super-acidic capsid surface containing 8,370 additional negative charges per capsid (totally 15,199 net negative charges in 9DE-T4 compared to 6,829 in WT-T4). The 9DE-T4 heads bound strongly to anion-exchange columns requiring a much higher NaCl concentration to elute, and a slightly larger 9DE-gp23* band was also visualized by SDS-PAGE (Fig. 6e I, II).

A head-to-head comparison of WT-T4-AVVs and 9DE-T4-AVVs showed that packaging efficiency was not affected by acidification. Like the WT head, the 9DE-head packaged ~8 molecules of 6.3 Kbp *Luci* plasmid per head (Fig. 6e III). Remarkably, however, head to head comparison of the AVVs assembled with the super-acidic heads exhibited ~5-fold greater transduction efficiency than the WT-AVVs, as measured by the delivered luciferase activity at MOIs of $10^3$ and $10^4$ (Fig. 6f). The enhanced delivery was further confirmed using a second *GFP* reporter plasmid control measured as percentage of cells showing green fluorescence (Fig. 6g). Furthermore, the 9DE-T4 showed maximum delivery at an order of magnitude lower dose (MOI of $10^4$) than WT-T4, further eliminating the potential for dose toxicity by T4-AVVs. Importantly, the enhanced transduction efficiency of the super-acidic AVVs was also recapitulated for delivery and expression of the full-length *dystrophin* payload. As shown in Fig. 6h, the 9DE-T4-AVVs exhibited enhanced expression of *mCherry* transgene which is part of the 17 Kbp polygene expression cassette containing the full-length *dystrophin* gene (Fig. 2g). The expressed full-length dystrophin protein also showed a more intense 427 kDa dystrophin band in 9DE-T4-AVVs when compared to WT-T4-AVVs (Fig. 6i).

## Re-wiring the capsid interior

To further expand the programmability of T4-AVVs, we again used CRISPR to re-wire the interior space of capsid with packaged proteins in addition to DNAs (Fig. 7a). This would not only increase the cargo capacity but also impart a distinctive property to T4-AVVs. By packaging proteins that bind to DNA, assembly of DNA-protein complexes can form in situ with the packaged DNA that, after delivery, can guide the transport of DNA cargo to nucleus (Fig. 7b). Such a guided transport system (GTS) could be adapted for guiding cargos to appropriate intracellular destinations, another common problem in payload delivery.

To re-wire the interior and establish GTS, we packaged NLS-fused LacI repressor [LacI(NLS)] molecules inside the head nanoparticle by CRISPR engineering. These would then form complexes with the incoming DNA molecules during packaging that contain an engineered *lac* operator sequence (*LacO*) in *trans* (Fig. 7a, b). We first generated an acceptor phage by CRISPR by deleting *ipI* and *ipII* genes (*10-amber.13-amber.HocΔ.SocΔ.ipIΔ.ipIIΔ* T4) and used this phage to replace *ipIII* with *LacI* by homologous recombination (Supplementary Fig. 7d–f). The

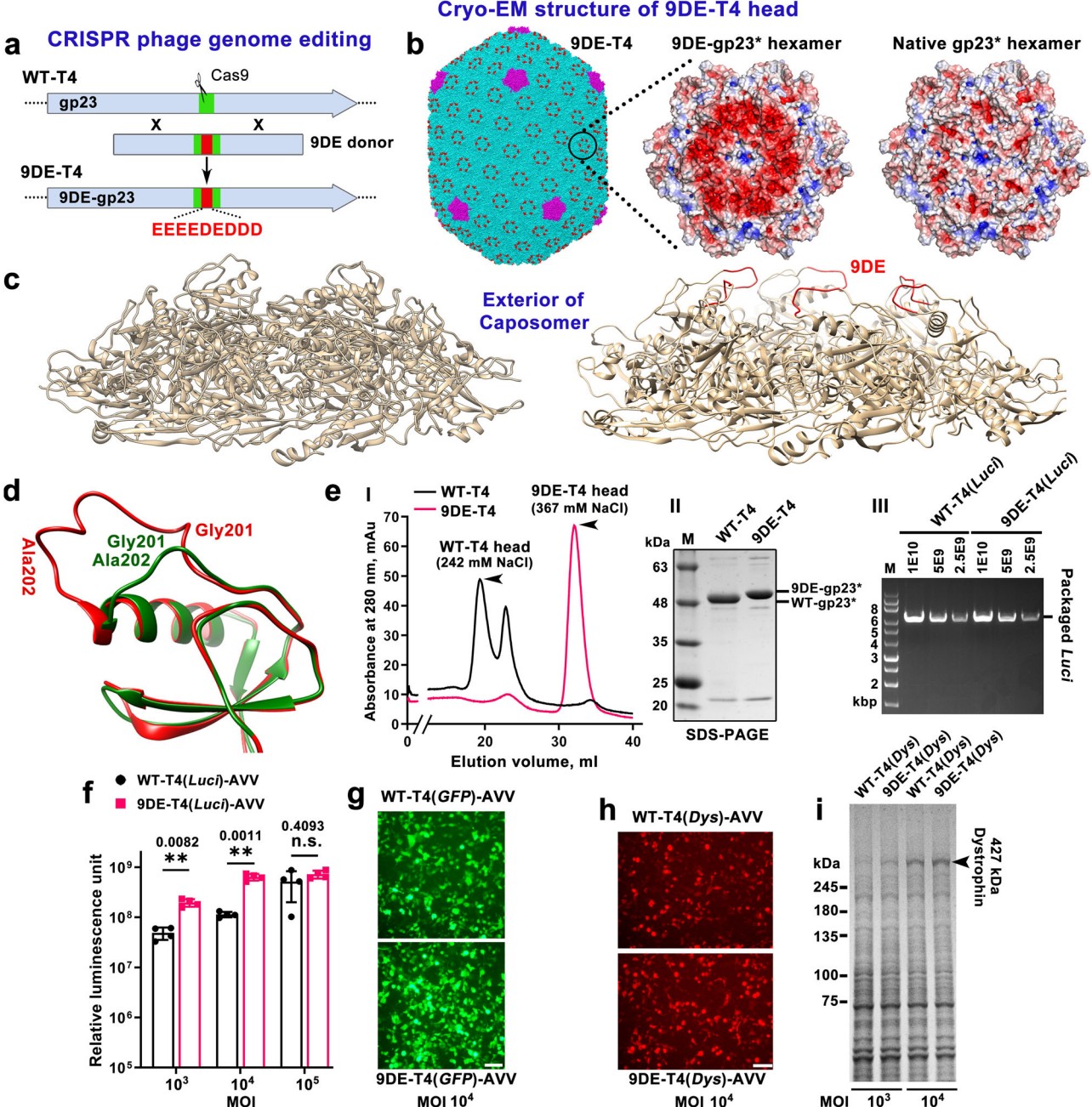

**Fig. 6 | The re-wired super-acidic T4 head showed enhanced cargo delivery.**
**a** CRISPR cleavage and homologous recombination between the T4 genome ends and the donor DNA (green) flanking the 9DE inset (red) resulted in the insertion of the 9DE sequence into the I domain loop (green) of the major capsid protein gene *gp23*. **b** High resolution (3.9 Å, near atomic level) cryo-EM structure of 9DE-T4 head (left). Six 9DE loops (red) in each gp23* hexamer (middle) create an acidic ring in each capsomer. Molecular surfaces of 9DE-gp23* capsomer (middle) and native gp23* capsomer[22] (PDB 7VS5) (right) are colored according to electrostatic potential. The color ranges from red, corresponding to a potential of −5 kT/e⁻, to blue, corresponding to a potential of +5 kT/e⁻. **c** Comparison of the WT (native)-gp23*[22] (left) and 9DE-gp23* (right) capsomer structures (side view). The 9DE insertion loops are shown in red. **d** The I domain 9DE insertion loop of mutant gp23* containing nine additional negatively-charged acidic amino acids (red) is superimposed onto the native structure[22] (forest green). **e** Characterization of 9DE-T4 heads. **I** Anion-exchange chromatography profiles of WT-T4 and 9DE-T4 heads. (**II**) SDS-PAGE showing the 9DE-gp23* band migrating slower than the WT gp23* due to its slightly larger size. (**III**) 9DE-T4 head and WT-T4 heads exhibit comparable in vitro DNA packaging efficiencies. **f** Transduction efficiencies of WT-T4(*Luci*)-AVVs and 9DE-T4(*Luci*)-AVVs as determined by luciferase activity at MOIs 10³, 10⁴, and 10⁵. Values represent mean with SD (*n* = 4). **P < 0.01, n.s., not significant. Paired *t* test (two-tailed) was used for comparison in each MOI. **g** Transduction efficiencies of WT-T4(*GFP*)-AVVs and 9DE-T4(*GFP*)-AVVs as determined by GFP expression at a MOI of 10⁴. Bar = 50 μm. **h, i** Transduction efficiencies of WT-T4(*Dys*)-AVVs and 9DE-T4(*Dys*)-AVVs at a MOI of 10⁴. The mCherry reporter (**h**) and 427 kDa dystrophin (**i**) genes are co-expressed as part of the 17 Kbp polygene expression cassette (see Fig. 2g). Bar = 50 μm.

N-terminus of LacI was fused to a 10-amino acid capsid targeting sequence (CTS) of IPIII that would localize the CTS-LacI in the inner scaffolding core of T4 capsid[49,50]. The CTS will then be cleaved off by the T4 prohead protease gp21 during head maturation reactions leaving LacI in the core while the CTS peptide escapes the capsid[1,38,49].

The rescued recombinant phage is devoid of all three IPs but would contain the packaged LacI protein molecules in their place. Indeed, our data demonstrate that the empty heads prepared from this mutant phage (*10-amber.13-amber.HocΔ.SocΔ.iplΔ.ipllΔ.ipIIIΔ.LacI* T4) contained ~370 molecules of LacI(NLS) inside the shell and showed

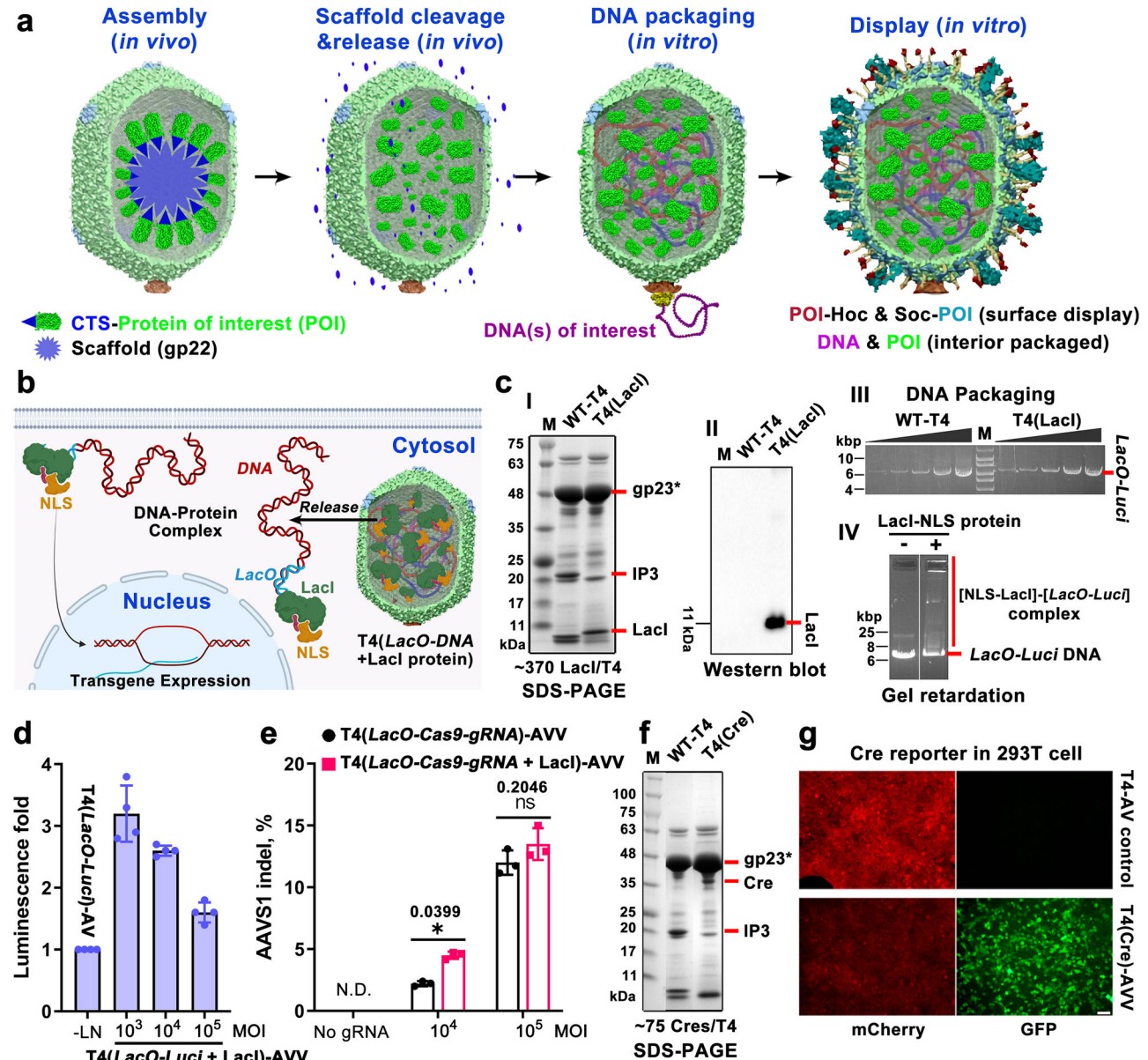

**Fig. 7 | A CRISPR re-wired guided transport system (GTS) installed in the T4-AVVs. a** Schematic illustration of sequential assembly to generate protein- and DNA-packaged T4-AVVs. The gene for the protein-of-interest (POI) is inserted into T4 genome by replacing the non-essential *ipl*, *ipII*, and ip*III* genes by CRISPR engineering. The 10-aa capsid targeting sequence (CTS) at the N terminus of POI leads to the packaging of POI into T4 heads as part of the scaffolding core during in vivo (*E.coli*) assembly. The POI remains in the capsid following CTS cleavage and head maturation. These heads are then used to package DNAs inside and display additional POIs on the exterior. **b** Schematic showing intracellular delivery of T4[*LacO-DNA* + LacI-NLS protein]-AVVs. The NLS-LacI-*LacO-DNA* complex delivered by the AVVs is transported into nucleus for transgene expression. **c** Functional characterization of T4(LacI protein)-heads. SDS-PAGE (**I**) and Western blotting (**II**) show

the T4 head-packaged LacI-NLS protein. (**III**) The T4(LacI)-heads exhibit comparable in vitro DNA packaging activity as the WT heads; (**IV**) binding of LacI-NLS protein to LacO-containing plasmid DNA as determined by gel retardation assay. **d** *LacO-Luci* DNA delivery is enhanced by T4(*LacO-Luci* + LacI-NLS protein)-AVVs compared with T4(*LacO-Luci*)-AVVs (*n* = 4). **e** Quantification of AAVS1 indel efficiency shows that T4(*LacO-Cas9-gRNA* + LacI-NLS protein)-AVVs enhance genome editing compared to AVVs lacking LacI-NLS protein (*n* = 3). **f, g** Characterization of Cre-packaged AVVs. SDS-PAGE of Cre-heads showing the presence of Cre in the purified heads (**f**). Site-specific recombination by Cre following delivery of T4(Cre protein)-AVVs into Cre-reporter cells (**g**). Bar = 50 μm. Values represent mean with SD. N.D., not detected. *$P < 0.05$, n.s., not significant, two-way ANOVA test.

comparable in vitro DNA packaging efficiencies as the WT heads (Fig. 7c I to III).

Next, we inserted the *LacO* sequence into the *Luci* or *Cas9-gRNA* plasmid and packaged it into the T4(LacI protein) capsids. The packaged LacI repressor and *LacO-DNA* then form DNA-protein complexes as seen in in vitro gel retardation experiments (Fig. 7c IV). These AVVs upon transduction into human cells showed enhanced expression of luciferase up to ~3.5 fold (Fig. 7d) and Cas9-mediated genome editing by ~2 fold (Fig. 7e), presumably through enhanced transport of DNA-

LacI(NLS) complex to nucleus due to the NLS signal (Fig. 7b). The observed enhancement was most significant when the MOI was low, ~$10^3$–$10^4$ AVVs per cell but not at a high ~$10^5$:1 ratio. This was probably because the delivery of more copies of DNA at a high MOI compensated for the enhanced LacI-mediated transport at a low MOI (Fig. 7d, e).

To test if the GTS strategy could be generally applied to DNA binding proteins that could carry out other genome modifications, we packaged Cre recombinase molecules using the same strategy (Fig. 7f).

In addition, we have also inserted reporter genes for *GFP* and *β-galactosidase* that could be generally useful for AVV engineering (Supplementary Fig. 7g, h). All these proteins were successfully packaged into T4 capsids although the copy number varied probably due to differences in size and structure of the protein that might affect incorporation into the scaffolding core. However, all the packaged proteins retained their biological functions; for instance, AVVs packaged with GFP protein and *mCherry* plasmid DNA exhibited both green and red fluorescence upon delivery, the former delivered as protein (green fluorescence phage) while the latter was expressed from delivered DNA (Supplementary Fig. 7g). Similarly, the packaged β-galactosidase formed functional tetramers and generated the blue phage due to its glucosidase activity (Supplementary Fig. 7h). Finally, AVVs containing packaged Cre recombinase carried out recombination between the LoxP sites of the *LoxP-mCherry-stop-LoxP-GFP* cassette resulting in splicing of the *mCherry* gene which in turn allowed expression of the *GFP* gene from an upstream promoter of the spliced product in Cre-reporter 293T cells (Fig. 7g).

## Discussion

The ability to assemble virus mimics that can be directed to perform defined molecular operations in human cells remained as the holy grail of medicine[6,11,83]. Here, we report proof for such a concept. We describe a sequential assembly-line approach to build AVVs in the test tube using purified and well-characterized structural components of bacteriophage T4, each engineered to perform a specific task(s) in a human cell (Supplementary Movie 1). These include attachment and entry into cells, intracellular trafficking, nuclear localization, and genome remodeling. In addition to creating enormous engineering space, this assembly-line approach allows mixing and matching of the components in desired combinations to generate varieties of AVVs endowed with specific therapeutic capabilities. Several features of this custom-buildable, plug-and-play T4-AVV platform distinguish it from the traditional phage and viral delivery platforms[12,84–86].

One of the powerful features of the T4-AVV platform is its ability to incorporate many types of therapeutic biomolecules including DNAs, proteins, RNAs, and their complexes in different compartments of the nanoparticle structure. These molecules, upon delivery into a human cell, faithfully execute their function(s) either independently or through interactions with another component(s). This has been demonstrated across a wide spectrum of molecules; proteins ranging from 27 kDa GFP to 516 kDa tetrameric β-galactosidase enzyme, nucleic acids ranging from ≤100-nt siRNAs and gRNAs to 20-40 kbp plasmid/viral DNAs, and preformed complexes including protein-protein, RNA-protein, and DNA-protein complexes. Functional circuits form between delivered molecules upon AVV transduction that can also be tunable by adjusting the copy numbers of the cargo molecules, providing many options to create AVVs with therapeutic potential.

Consistently high efficiency of T4-AVV delivery into cultured human cells was measured either by the expression of a reporter gene (e.g., *Luci*, *GFP*, and *mCherry*) or by the functional activity of a delivered protein (e.g., GFP, β-Gal, Cre, and Cas9). A critical component that contributed to the high efficiency of the AVVs is the lipid coat, which we created by taking advantage of the highly anionic character of the T4 capsid surface[21,22]. The cationic lipids spontaneously bound to T4 capsid generating a lipophilic and cationic coating that, overall, is complementary to the anionic surface of human cells[61,77,87,88]. Without this coat, the transduction efficiency was poor. Even the particles that are cationic (TAT displayed) but lacking the lipid coat showed a much lower signal. Furthermore, the lipid coat did not impair the display or functionality of Soc- and Hoc-fused molecules and complexes. On the other hand, these molecules, particularly the positively charged ones, further accentuated the T4-AVV transduction efficiency.

The T4-AVVs broke through five significant barriers that currently exist for the delivery of biomolecules into human cells. First, the T4-

AVVs can efficiently deliver multiple copies of multiple and relatively large DNA molecules into cells. This has been amply demonstrated using a series of plasmids containing reporter genes, functional genes, genome editing genes, and even a viral genome. Consequently, the reporter signal as measured by luciferase activity is high, even higher than AAV2 transduction which due to its small cargo capacity is limited to one reporter molecule per particle. Perhaps the most significant, from a therapy standpoint, is the delivery of the full-length *dystrophin* gene that resulted in the expression of the ~427 kDa dystrophin protein in transduced cells. This is possible not only because of the large cargo capacity of T4 but also because of the promiscuous nature of T4's packaging machinery that exhibits no sequence dependence[51,60,89]. Additionally, the encapsidated DNA remained stable as it is confined within the capsid shell. The only pore through which the packaged DNA has any possibility of leaking out is the portal ring[23,44]. At the most constricted region of the ring, the ~14-23 Å internal diameter of the pore is insufficient for the ~23 Å-diameter DNA double helix to translocate passively[25]. Furthermore, our assays showed that the T4-AVVs are stable at 4 °C for at least four weeks showing no significant loss of DNA, and no obvious leakage of DNA from packaged heads was evident in EM images.

The second barrier that the T4-AVVs broke through is the all-in-one delivery. As demonstrated, a series of T4-AVVs were assembled that efficiently delivered complex cargos consisting of combinations of DNAs, proteins, RNAs, and their complexes. This is essential for many genome remodeling applications including genome editing and gene recombination that require co-delivery of multiple biomolecules[68,70,90]. For example, for genome editing, we assembled AVVs in five different all-in-one configurations carrying Cas9 nuclease and gRNAs either as functional RNP complexes displayed outside and/or as expressible genes packaged inside. Similarly, a series of gene recombination AVVs were assembled that co-delivered the donor plasmid and the site-specific recombinase Cre. Such all-in-one multicomponent deliveries will in the future allow the design of therapeutic payloads for multigenic and multifactorial diseases including many cancers, neurodegenerative disorders, and cardiovascular diseases.

The third barrier that the T4-AVVs broke through is multiplex delivery. We were able to assemble T4-AVVs by incorporating cargo molecules not only to target multiple sites (e.g., multiple gRNAs and siRNAs) but also to perform multiple molecular operations in the human genome. In one combination, three different operations; genome editing, gene expression, and site-specific recombination were simultaneously performed by incorporating Cas9 and gRNA as displayed RNPs, *GFP* or *Luci* genes as packaged plasmids, and Cre recombinase and donor plasmid as displayed and packaged molecules, respectively. In another combination, gene silencing, gene expression, and genome editing were simultaneously performed by incorporating siRNAs, mRNA, Cas9, and gRNAs into the same AVV.

The fourth barrier that the T4-AVVs broke through is programmability, the ability to execute a set of instructions as well as generate functional circuits involving the delivered molecules. Many examples cited above demonstrated the functional outcomes that each AVV is programmed with. In addition, functional circuits formed upon entry, as demonstrated by the repurposing of Cas9 function. In one configuration, Cas9 delivered the bound siRNA and then, through interaction with the gRNA expressed from a co-delivered plasmid, it formed an editing complex, which then carried out genome editing at the targeted site. This illustrates how three independent operations; gene silencing, gene expression, and genome editing, can be programmed into one payload.

Finally, we demonstrated how the nanoparticle itself can be re-wired to enhance the programmability of T4-AVVs. Hundreds of peptides and/or protein molecules were incorporated into both the inside and the outside of the nanoparticle structure by CRISPR engineering.

Consequently, the re-wired AVV shell has 930 additional peptides carrying 8,370 negative charges on its surface and/or ~270 molecules of site-specific DNA binding protein LacI inside it. These AVVs are now endowed with functions such as guidance systems for intracellular trafficking. Besides creating enormous engineering space, such re-wiring allows for unique functionalizations of AVVs for targeted therapies.

In conclusion, we have created an additional category of nano-material, a bacteriophage-based artificial viral platform, which can be custom-produced in the test tube using an assembly-line approach (Supplementary Movie 1). These AVVs possess a similar architecture as natural viruses, though the mechanisms of entry, endosomal escape, disassembly, and intracellular trafficking require detailed investigations for maximizing delivery[83,91]. Virtually unlimited varieties of AVVs can be assembled using this approach that can faithfully execute functions each is programmed with and make precise alterations in the human genome and cellular metabolism. Their large cargo capacity, ability to incorporate diverse cargos, programmability, and all-in-one delivery provide hitherto unavailable incentives for designing therapies for many human diseases and rare disorders. Furthermore, the T4-AVVs can be produced at low-cost and high yield using simple bacterial infection (~$10^{14}$ particles from 1 L of *E. coli* culture) and the head nanoparticles are stable for several months at 4 °C or at −70 °C. This platform is now poised for therapeutic applications, correction of defects in primary human cells, both ex vivo and in vivo. However, it would require specific payload designs by traditional as well as CRISPR engineering, as well as leveraging the emerging lipid nanoparticle technologies[77]. Additionally, safety concerns may arise when T4-AVVs are transitioned to the clinic, such as potential unwanted responses by the host immune system or off-target effects[92]. These are currently under investigation to transition the T4-AVV platform from bench to bedside.

## Methods

### Sequence Information
The sequences of DNA templates, gRNA sequences, primers, and Hoc- or Soc- fused recombinant proteins are available in Supplementary Note 1.

### Recombinant Protein Expression and Purification
Recombinant proteins (except Cas9 and Cpf1) were expressed by transforming the pET28b expressing plasmid into *Escherichia coli* (*E. coli*) BL21 (DE3) RIPL cells by the heat-shock method. The transformed cells were grown to an OD600 of 0.5 at 37 °C in Moore's medium (20 g of tryptone, 15 g of yeast extract, 8 g of NaCl, 2 g of dextrose, 2 g of $Na_2HPO_4$, and 1 g of $KH_2PO_4$ dissolved in 1 L of Milli-Q water) containing 50 µg/ml kanamycin and 25 µg/ml chloramphenicol, and protein expression was induced by 1 mM isopropyl β-d-1-thiogalactopyranoside (IPTG) at 30 °C for 3 h. After induction, cells were harvested by centrifugation, and the pellets were suspended in binding buffer (50 mM Tris-HCl, 300 mM NaCl and 20 mM imidazole, pH 8.0) containing proteinase inhibitor cocktail (Roche, USA) and benzonase nuclease (Millipore Sigma). The cell suspension was lysed by French press (Aminco), and the soluble fraction was separated from cellular debris by centrifugation at $34,000 \times g$ for 30 min. The lysate was filtered through 0.22-micron filters (Millipore, Stericup) and applied to a pre-equilibrated (binding buffer) HisTrapHP column (AKTA-Prime, GE Healthcare) and washed with binding buffer. The His-tagged protein was then eluted with a 20–500 mM linear imidazole gradient. The peak fractions were further purified by size-exclusion chromatography using the Hi-Load 16/60 Superdex-200 (prep-grade) gel filtration column (GE Healthcare) in GF buffer (20 mM Tris-HCl and 100 mM NaCl, pH 8.0) according to the manufacturer's instructions. The fractions containing the desired protein were pooled and concentrated by AmiconUltra-4 centrifugal filtration (10 kDa cut-off;

Millipore), flash-frozen in liquid nitrogen, and stored at −80 °C. All the column operations were performed at 4 °C. Gel filtration molecular size standards were chromatographed on the same column to calculate the approximate size of the purified protein.

For Cas9-Soc or Cpf1-Soc purification, the recombinant SpCas9 or LbCpf1 used in this study was fused to Soc at the C-terminus and to nuclear localization signal peptide at the N-terminus. The protein also has a C-terminal hexa-histidine tag. Briefly, *E. coli* RIPL cells were grown at 37 °C until OD600 = 0.6 and incubated at 20 °C for 40 min, then induced with 0.25 mM IPTG. After 20 h, the cells were collected and resuspended in 50 ml of binding buffer (50 mM Tris-HCl, 300 mM NaCl, 20 mM imidazole, and 5 mM Tris (2-carboxyethyl) phosphine (TCEP, pH 8.0) (Soltec Ventures) containing protease inhibitor cocktail (Roche, USA) and benzonase (Millipore Sigma). The Cas9-Soc or Cpf1-Soc proteins were then purified by HisTrapHP and Superdex-200 columns as described above.

### T4 CRISPR Engineering
T4 phage engineering was performed according to a previously described procedure[54,93]. A brief description is as follows. The *E. coli* strains P301 (*sup^0*) and B40 (*sup^1*) were used as host strains for plasmid transformations. The *10-amber 13-amber hoc-del soc-del* T4 phage was propagated on *E. coli* B40 (*sup^1*). CRISPR-Cas9 or Cpf1 plasmids with specific spacer(s) were constructed by cloning spacer sequences into the streptomycin-resistant plasmid DS-SPCas (Addgene no. 48645). Sequences of the spacers of *gp23*, *ipI*, *ipII*, and *ipIII* genes are shown in Supplementary Note 1. The homologous donor plasmids were constructed by cloning the donor DNA into the pET28b vector. The CRISPR-Cas9/Cpf1 and donor plasmids were co-transformed into amber-suppressor-containing B40 cells, and then the positive clones were selected by streptomycin and kanamycin antibiotics. The cells transformed with either the CRISPR-Cas9/Cpf1 plasmid or the donor plasmid were used as controls. The cells were infected with WT or *10-amber 13-amber hoc-del soc-del* T4 phages to produce the first-generation (G1) phage plaques. The diluted G1 phages were applied to infect spacer-containing B40 cells to produce G2 progenies. This efficiently eliminated any parental phage background under CRISPR pressure. Single G2 plaques were picked and used to infect B40 cells (without spacer or donor) to produce purified G3 phages. The engineered genome of the G3 plaque was amplified and sequenced to confirm the insertion or deletion.

### T4 heads purification
The 10-amber 13-amber hoc-del soc-del T4, 10-amber 13-amber hoc-del soc-del ipI-del ipII-del ipIII-del CTS-LacI (or foreign gene) T4 (LacI or foreign protein-packaged), or 9DE-gp23 10-amber 13-amber hoc-del soc-del T4 (9DE-T4) phages were propagated on E. coli B40 (sup¹, with amber-suppressor). Then, E. coli P301 (sup-, no amber-suppressor) cells (500 ml of culture) were infected with the above mutant phages at an MOI of 4. After 5 min, the culture was infected again at the same MOI, followed by 30 min shaking-incubation at 37 °C. The culture was centrifuged at $25,000 \times g$ for 45 min and the pellet was resuspended in 40 ml of Pi-Mg buffer (26 mM $Na_2HPO_4$, 68 mM NaCl, 22 mM $KH_2PO_4$, and 1 mM $MgSO_4$, pH 7.5) supplemented with 4 µl benzonase (Millipore Sigma) to digest the DNA. One ml chloroform was added to lyse the T4 head-containing cells, followed by incubation at 37 °C for 30 min. After two rounds of low-speed ($4000 \times g$ for 10 min) and high-speed ($25,000 \times g$ for 45 min) centrifugation, the pellet was resuspended in 200 µl of Tris·Mg buffer (10 mM Tris-HCl, 50 mM NaCl, and 5 mM $MgCl_2$, pH 7.5), followed by CsCl density gradient centrifugation. The extracted T4 heads were dialyzed overnight against Tris·Mg buffer and further purified by anion-exchange chromatography (diethylaminoethanol (DEAE) -Sepharose or quaternary ammonium (Q) -Sepharose)[94]. The peak capsid fractions were concentrated and stored at −80 °C. The number of particles was determined by quantification of

the major capsid protein gp23* (48.7 kDa, -0.75 µg gp23 per $10^{10}$ T4 heads) in comparison with the known amounts of phage T4 or the bovine serum albumin (BSA) protein standard (Thermo Scientific), using SDS-PAGE and laser densitometry. Prep-to-prep variation of the purified T4 head particles used for DNA packaging or protein display is small and any preparation that showed poor efficiencies (for whatever reason) was discarded.

## T4-AVV assembly

DNA Packaging: for a 20 µl packaging reaction mixture, the packaging buffer (30 mM Tris-HCl, 100 mM NaCl, 3 mM $MgCl_2$, and 1 mM ATP, pH 7.5), the linearized plasmid DNA, and the purified full-length gp17 (~3 µM) were sequentially added and incubated at 37 °C for 5 min, followed by the addition of $2 \times 10^{10}$ T4 heads. The mixture was further incubated at 37 °C for 30 min, followed by the addition of 0.5 µl ben-zonase nuclease and incubation at 37 °C for 20 min to remove excess unpackaged DNA. The packaged nuclease-resistant DNAs were released by treatment with 0.5 µg/µl proteinase K (Thermo Scientific), 50 mM ethylenediaminetetraacetic acid (EDTA), and 0.2% SDS for 20 min at 65 °C. The packaged DNA was analyzed using 1% (wt/vol) agarose gel electrophoresis. The amount of packaged DNA was quantified by Quantity One software (Bio-Rad) and the known con-centration of DNA marker (GoldBio) was used as the standard. The packaging efficiency was defined as the number of DNA molecules packaged in one T4 head on average.

Protein and RNA Display: After packaging linearized DNA as above, Soc- and/or Hoc-fused proteins were added to the packaging mixture at different ratios and incubated at 4 °C for 1 h. The mixtures were sedimented by centrifugation at $15,000 \times g$ for 1 h, and unbound proteins in the supernatants were removed. After washing twice with 1X PBS (pH7.4), the pellets were resuspended in PBS (pH7.4) for SDS-PAGE analysis or Opti-MEM for cell transduction. After Coomassie Blue R250 (Bio-Rad) staining and destaining, the protein bands on SDS-PAGE gels were quantified by laser densitometry (PDSI, GE Healthcare). The densities of Hoc fusion, Soc fusion, and gp23* bands in each lane were quantified independently, and the copy numbers of bound Hoc or Soc fusion molecules per T4 were calculated using gp23* band (48.7 kDa) in each lane as the internal control (930 copies per T4 capsid, -0.75 µg gp23 per $10^{10}$ T4 heads). For gRNA/siRNA/mRNA dis-play, T4 heads displayed with Hoc or Soc fusion protein molecules were resuspended in RNAase-free PBS (pH7.4) buffer, and then incu-bated with RNA at 4 °C for 1 h. The T4-RNP complexes were sedi-mented by centrifugation at $15,000 \times g$ for 1 h, and unbound RNAs in the supernatants were removed. After washing twice with PBS (pH7.4), the pellets were resuspended in Opti-MEM for transduction. To quantify the binding of RNA, the T4-RNP complex was treated with 0.5 µg/µl proteinase K (Thermo Scientific), 50 mM ethylenediamine-tetraacetic acid (EDTA), and 0.2% SDS for 30 min at 65 °C to release the packaged DNA and displayed RNA, followed by agarose gel electrophoresis.

To compare the delivery efficiency among T4-AVVs displayed with various proteins or RNA-protein complexes (RNPs) (Figs. 3–5), we used the same batch of purified T4 heads (-$10^{14}$ T4 heads from 1 L culture) for the entire experiment and conducted bulk DNA packaging in a single Eppendorf tube. Then, the packaged heads were equally divided for the display of various proteins or RNPs (in the same reaction mixture). The number of head particles was determined by SDS-PAGE (0.75 µg gp23 per $10^{10}$ T4 head particles). Thus, each tube used for display received the same number of head particles packaged with the same amount of DNA as quantified by agarose gel electrophoresis. The copy numbers of proteins or RNPs displayed in each case were then determined by SDS-PAGE and/or agarose gel electrophoresis using the appropriate internal controls described above. The protein band intensity was quantified by Image J software (National Institutes of Health).

## Lipid coating

For 24-well plate transduction, the DNA-packaged and/or protein-displayed T4 nanoparticles as above were diluted in 50 µl of Opti-MEM and mixed gently. Meanwhile, 50 µl Opti-MEM medium was added to a separate sterile tube, followed by the addition of an appropriate amount of cationic lipids such as lipofectamine 2000, lipofectamine 3000, lipofectamine RNAiMAX, lipofectamine LTX, lipofectamine stem, and ExpiFectamine 293 (EXPI) (Thermo Scientific). After 5 min incubation, the T4 particles were added, gently mixed, and incubated for 20 min at room temperature without shaking. The total volume of the mixture is 100 µl.

The T4-AVVs and the controls thus assembled were used for all the transduction and delivery experiments. During preparation and cell transduction stages, T4-AVVs are handled under standard laboratory conditions, which are equivalent to biosafety level 2. These conditions include standard safety protocols such as the use of appropriate per-sonal protective equipment and maintaining aseptic conditions to minimize the risk of contamination.

## Cell culture

HEK293T cells were cultured in Dulbecco's Modified Eagle's Medium (DMEM, Gibco) supplemented with 10% fetal bovine serum (FBS, Gibco), 1 × HEPES (Gibco), and 1% antibiotics (Gibco) (complete DMEM). Cells were maintained in a humidified atmosphere at 37 °C and 5% CO2 and grown until -80–90% confluency. Cells were then dissociated from adherent surfaces using 0.05% trypsin/EDTA (Gibco) and passaged at a subcultivation ratio of 1:5.

## Cell Transduction

One day before transduction, HEK293T cells were transferred to 24-well plate at $2 \times 10^{5}$ cells per well in complete DMEM. On the day of transduction, the cells were incubated with T4-AVVs in antibiotic-free Opti-MEM for 6 h. About $2 \times 10^{10}$ particles were transduced into cells in each well (MOI $10^{5}$). This optimal condition applies to most of the experiments reported in this study unless stated otherwise. Thereafter, the Opti-MEM was removed and replaced with complete DMEM. The cells were further incubated at 37 °C for an additional 48 h for further analysis. GFP/mCherry transgene expression was observed by fluor-escence microscopy (Carl Zeiss) at 48 h post-transduction, and the average fluorescence intensities were quantified by ImageJ software (1.8.0_172, NIH). The nucleus was counterstained with Hoechst 33342 (Thermo Scientific) at 37 °C for 20 min. The number of Hoechst-stained and GFP-Hoechst double-stained cells in three fields of the microscope were counted to approximately determine the effi-ciency of transduction. This was calculated as the number of GFP and Hoechst double-stained cells divided by the number of Hoechst-stained cells (or percentage of cells expressing GFP fluorescence).

## Mass Spectrometry

HEK293T cells were transduced with T4(*Dys*)-AVVs and vector control as described in the Results. Cells were incubated at 37 °C for 72 h post-transduction. M-PER Mammalian Protein Extraction Reagent (Thermo Scientific) was used to extract the cytoplasmic and nuclear protein from the transduced 293T cells according to the manufacturer's instructions. Briefly, the culture medium was carefully removed and the cells were washed twice with PBS (pH7.4). Approximate 200 µl M-PER reagent was added to each well of the 24-well plate, followed by gently shaking for 5 min. The lysate was collected and transferred to a centrifuge tube and centrifuged at $14,000 \times g$ for 10 min. The super-natant was transferred to a new tube for 4-20% SDS-PAGE analysis.

Approximately 10 µg cell protein extract was loaded into each gel lane, followed by electrophoresis at a constant voltage of 150 V for 6 h. The gel was carefully removed from the holder and stained by Coo-massie brilliant blue R-250. Then the gel was washed with water twice (10 min each). The dystrophin band from T4(*Dys*)-AVVs sample and the

corresponding region from the vector control sample were excised from the gel. The excised piece was sliced into 1 mm cubes and transferred to a 1.5 ml microfuge tube. The samples were analyzed by the Taplin Biological Mass Spectrometry Facility (Harvard Medical School, Boston, MA) to generate peptides and their sequence determination by microcapillary LC/MS/MS tools.

### AAV2 production, purification, and quantification

The single-stranded rAAV2 (ITR-CMV enhancer and promoter-fireflyLuci-hGH poly(A)) particles were prepared, or obtained from Dr. Mavis Agbandje-McKenna's lab by using the standard triple transfection protocol[95]. Briefly, rAAV2 viruses were generated by triple transfection of HEK293T cells with the adenoviral helper plasmid (pAAV Helper), a rep/cap plasmid expressing rep and cap proteins (pAAV Rep-Cap), and an inverted terminal repeat plasmid containing luciferase (pAAV Luciferase). The three plasmids were cotransfected at a 1:1:1 ratio into HEK293T cells expressing adenovirus E1a&E1b proteins using polyethyleneimine (Polysciences, PA). After 5 days, the cells were harvested and resuspended in lysis buffer [50 mM Tris-HCl (pH 8.5) and 0.15 M NaCl], and then lysed by three freeze/thaw cycles in liquid-nitrogen and a 37 °C water bath. Benzonase was added to the mixture at 50 U/ml, and the lysate was incubated at 37 °C for 30 min. The lysate was clarified by centrifugation at 3700 g for 20 min, and the virus-containing supernatant was filtered through a 0.22 μm polyethersulfone membrane (crude lysate).

Affinity chromatography was performed using an AKTA Prime chromatography system equipped with a HiTrap AVB Sepharose HP column (GE Healthcare, IL) according to the manufacturer's instructions. The crude lysate was loaded in the column equilibrated with PBS (pH 7.4) at a flow rate of 2 ml/min. After loading, the column was washed with PBS (pH 7.4) until the UV absorbance reached the baseline to remove contaminant proteins. Bound viruses were then eluted with 50 mmol/l glycine–HCl (pH 2.7) elution buffer. One milliliter elution fractions were collected into tubes containing 100 μl of 1 mol/L Tris–HCl (pH 8.0) to neutralize the low pH buffer. The collected AAV particles were pooled and concentrated and stored at −80 °C. The rAAV2 titers (viral genome-containing particles, vg) were determined by quantitative real-time PCR[95].

### Immunofluorescence staining

The cells transduced with T4(*Dys*)-AVVs or vector control were fixed using 4% paraformaldehyde in PBS (pH 7.4) for 10 min at room temperature. Then the samples were incubated for 10 min with PBS (pH 7.4) containing 0.25% Triton X-100 for permeabilization, followed by washing cells in PBS (pH 7.4) three times for 5 min each. The cells were blocked with 1% BSA, 22.52 mg/mL glycine in PBST (PBS + 0.1% Tween 20) for 30 min to prevent nonspecific binding of the antibodies. Then the cells were incubated with 1/500 diluted rabbit anti-firefly luciferase antibody in 1% BSA in PBST overnight at 4 °C. The cells were washed three times in PBS (pH 7.4), 5 min each wash, followed by incubation in the Alexa488-labeled goat-anti-rabbit secondary antibody with 1/500 diluted in 1% BSA for 1 h at room temperature in the dark. Finally, the cells were washed three times with PBS (pH 7.4) for 5 min each in the dark, stained the nucleus with Hoechst, and observed under fluorescent microscopy.

### Stable SARS-CoV2 Spike Trimer-producing Cell Line Construction through T4-AVV delivery

The CoV2 Spike trimer purification and characterizations were performed according to the protocols described previously[38]. Briefly, the culture supernatant of *Pac-CMV-S-ecto* stable cell line constructed by T4-AVV delivery was clarified through a 0.22-mm filter and then loaded on a HisTrap HP column (Cytiva) equilibrated with wash buffer [50 mM Tris-HCl (pH 8.0), 300 mM NaCl, and 20 mM imidazole] using AKTA-prime chromatography system (GE Healthcare). The protein-bound column was washed with wash buffer until the UV absorbance reached the baseline to remove contaminant proteins. The S trimers were eluted using a 20 to 300 mM linear gradient of imidazole. The peak fractions of S trimer were pooled and applied to a HiLoad 16/600 Superdex 200 (preparation grade) size-exclusion column (GE Healthcare) equilibrated with the gel filtration buffer [50 mM Tris-HCl (pH 8.0) and 150 mM NaCl] using the AKTA FPLC system. The eluted fractions were collected and stored at −80 °C until use.

Enzyme-linked immunosorbent assays (ELISAs) were performed to analyze the binding of the purified S-trimer to human ACE2 protein. Briefly, 100 ng of S-trimer protein was coated on plates overnight at 4 °C. After blocking with PBS–5% BSA buffer at 37 °C for 1 h, human ACE2-mouse Fc fusion protein (Sino Biological) was serially diluted and added to wells and incubated at 37 °C for 1 h. After washing 5 times with PBST (PBS + 0.1% Tween 20), plates were incubated with the goat anti-mouse IgG-HRP antibody (1/50,000) at 37 °C for 1 h. The plates were washed 5 times with PBST and developed with the TMB (3,3′,5,5′-tetramethylbenzidine) Microwell Peroxidase Substrate System (KPL). Reactions were stopped and the absorbance at 650 nm was measured on a VersaMax spectrophotometer.

### Quantification of luciferase activity

To analyze luciferase gene delivery into cells by T4-AVVs, luciferase activity was measured using the Promega (USA) Luciferase Assay System according to the manufacturer's recommended protocol. Briefly, the growth medium was removed, and cells were rinsed with PBS. After removing the wash buffer, 150 μl of passive lysis buffer was added to each well, followed by gentle shaking at RT for 20 min. Twenty microliters of the cell lysate were then transferred to a 96-well white opaque plate and mixed with 80 μl of Luciferase Assay Reagent, and the luminescence signal was recorded by the Glomax Multi Detection System (Promega). Triplicate measurements were applied to each group.

### Beta galactosidase (β-gal) transduction

The activity of the Soc-β-gal enzyme delivered by T4-AVVs into cells was determined by staining cells with X-Gal using the β-Galactosidase Staining kit (Sigma) according to the manufacturer's protocol. The representative staining images were captured by ChemiDoc Imaging System (Bio-Rad).

### Cell proliferation assay

Cell viability was determined using the CellTiter-Glo Luminescent Cell Viability Assay Kit (Promega) after transfection for 48 h following the manufacturer's protocol. Briefly, an equal volume of CellTiter-Glo Reagent was added to the cell culture in each well. The mixture was shaken for 2 min to induce cell lysis and then incubated at room temperature for 10 min to stabilize the luminescence signal, which was then recorded by the Glomax Multi Detection System (Promega). The viability of the untreated cell group was normalized to 100%, and triplicate measurements were applied to each sample.

### Western Blotting Analyses

Briefly, the T4 head sample or the extracted cellular protein sample was added with loading buffer and boiled for 10 min, separated by 12% SDS-PAGE, and transferred to nitrocellulose membranes (Bio-Rad). Blocking was performed in 5% BSA/PBS-T buffer (PBS with 0.05% Tween-20, pH 7.4) at room temperature for 1 h with gentle shaking. Blots were then washed five times with PBS-T. Primary anti-GFP (1/2,000), anti-tubulin (1/5,000), anti-dystrophin (1/100), or anti-His6 (1/2,000) antibodies were added to the blots and incubated overnight at 4 °C in PBS with 5% BSA. After washing with PBS-T five times, a secondary goat anti-mouse or goat anti-rabbit HRP-conjugated antibody (Abcam) was added at a 1:10,000 dilution in 5% BSA/PBS-T for 1 h at room temperature, followed by rinsing five times with PBS-T. Signals were visualized with an enhanced chemiluminescence substrate

(BioRad, USA) using the BioRad Gel Doc XR+ system and Image Lab software according to the manufacturer's instructions (BioRad, USA).

## Genomic DNA extraction and T7EI assay for genome editing
HEK293T cells were transfected with various genome editing AVVs as described in the Results. Cells were incubated at 37 °C for 72 h post-transduction. Genomic DNA was isolated using the GeneJET Genomic DNA Purification kit (Thermo Scientific) following the manufacturer's instructions. Briefly, cells were resuspended in a lysis solution/Proteinase K and incubated at 56 °C for 10 min, followed by the treatment with RNAase A at room temperature for 10 min. GeneJET column was used to absorb genomic DNA and washed with wash buffer. Genomic DNA was eluted with elution buffer and stored at −20 °C. The genomic region surrounding the AAVS1 or HBB target site was amplified, and PCR products were purified using QIAquick Gel Extraction Kit (Qiagen) following the manufacturer's protocol. A total of 400 ng or 200 ng of the purified PCR products were mixed with 2 μl 10X NEB buffer 2 (NEB) and nuclease-free water to a final volume of 20 μl, and annealed to enable heteroduplex formation using the following incubations: 95 °C for 10 min, 95 °C to 85 °C ramping at −2 °C/s, 85 °C to 25 °C at −0.1 °C/s, and 4 °C for hold. T7 Endonuclease I (NEB) was then added to the annealed PCR product and incubated at 37 °C for 30 min. T7EI digestion product was analyzed on 1.5% (wt/vol) agarose gel. Gels were imaged with a GelDoc gel imaging system (BioRad) and quantification was based on relative band intensities using ImageJ software. The estimated gene modification was calculated using the following formula: indel (%) = 100 x [1 − (1- fraction cleaved)$^{1/2}$].

## AAVS1 gRNA synthesis
A DNA template containing the T7 promoter, the gRNA target, and the gRNA scaffold sequences was amplified by PCR with Phusion High-Fidelity PCR Master Mix (Thermo Scientific). The T7-gRNA PCR fragment was gel-purified and used as a template for in vitro transcription using the HiScribe T7 High Yield RNA Synthesis Kit (NEB). T7 transcription was performed overnight, and the RNA was purified using the MEGAclear Transcription Clean-Up Kit (Thermo Scientific). The gRNA was eluted with RNase-free water, analyzed by agarose gel electrophoresis, quantified with Nanodrop 2000 (Thermo Scientific), and stored at −80 °C.

## CRISPR RNP binding and cleavage assay
To test the binding of Cas9 or Cas9-Soc to gRNA/siRNA/mRNA, the purified protein and RNA at different ratios were incubated at room temperature for 1 h, and then analyzed by agarose gel electrophoresis. The genomic region surrounding the AAVS1 target site was amplified by PCR with Hot-Start DNA Polymerases (Thermo Scientific), purified by QIAquick Gel Extraction Kit (Qiagen), and used as the substrate for Cas9 cleavage assay. In a reaction volume of 20 μl containing NEBuffer 3 (100 mM NaCl, 50 mM Tris-HCl, 10 mM MgCl$_2$, and 1 mM DTT, pH 7.9) and PCR product (300 ng), purified Cas9 or Cas9-Soc (50 nM) and AAVS1gRNA (50 nM) were added. After incubation for 1 h at 37 °C, the DNA was analyzed by 1.5% (wt/vol) agarose gel electrophoresis.

## Cre-Hoc recombination assay
LSL-GFP plasmid was used as the substrate for testing Cre-Hoc recombination in vitro. In a reaction volume of 50 μl containing recombination buffer (33 mM NaCl, 50 mM Tris-HCl, and 10 mM MgCl$_2$, pH 7.5) and LSL-GFP plasmid, increasing amounts of purified Cre-Hoc protein were added. After incubation at 37 °C for 30 min and then at 70 °C for 10 minutes, the DNA was analyzed by 0.8% (wt/vol) agarose gel electrophoresis.

## VRCO1 and CH58 antibody quantifications
HEK293T cells were transduced with T4-AVVs packaged with the linearized plasmids expressing the heavy and light chains of VRCO1 and/or CH58 antibodies. After culturing for 2 days, cell culture supernatants were harvested and analyzed for antibody production by ELISA. ELISA plates (Evergreen Scientific, 96-well) were coated with 0.1 μg of HIV-1JRFL gp140 envelope protein per well in coating buffer (0.05 M sodium carbonate-sodium bicarbonate, pH 9.6) overnight at 4 °C. After washing three times with PBS buffer (pH 7.4), the plates were blocked with PBS-5% BSA buffer for 1 h at 37 °C. Known quantities of purified VRCO1 or CH58 monoclonal antibodies in five-fold serial dilution were added to triplicate wells to generate a standard curve, with a starting concentration of 2000 ng mL$^{-1}$. The concentrations of VRCO1 or CH58 antibodies in cell culture medium were determined using a 5-fold dilution series in PBS-1%BSA. The diluted samples were added to each well, and the plates were incubated at 37 °C for 1 h and washed five times with PBS-T buffer (PBS with 0.05% Tween-20, pH 7.4). The secondary goat anti-human IgG-HRP antibody was then added to each well at a 1:5000 dilution and incubated for 1 h at 37 °C, followed by washing five times with PBS-T buffer. Next, the TMB (3,3′,5,5′-tetramethylbenzidine) Microwell Peroxidase Substrate System (KPL) was applied in the dark for color development. After 10 min, the enzymatic reaction was quenched by adding TMB BlueSTOP (KPL) solution, and plates were read within 30 min at 650 nm using an ELISA reader (VERSA max, Molecular Devices).

## Cryo-EM imaging and data processing
For the cryo-EM sample preparation, 3.5 μL aliquots of purified 9DE-T4 capsids were applied onto a glow-discharged 400-mesh copper grid with ultrathin carbon supported on a lacey film (Ted Pella, Catalog No. 01824). The grids were mounted onto the Vitrobot Mark IV instrument (Thermo Fisher Scientific), blotted for 4 seconds (with blot force 2), and plunged frozen in liquid ethane. The cryo-EM data were collected at Purdue University's Cryo-EM facility using a Titan Krios electron microscope (Thermo Fisher Scientific) operated at 300 kV. A total of 19,756 micrograph movies (each composed of 40 frames) were collected using the EPU software (Thermo Fisher Scientific) with a K3 Direct Detection Camera (Gatan) at a magnification of 64,000, resulting in the super-resolution pixel size of 0.666 Å. The total electron dose was 36 electrons/Å2. Motion correction to align the movie frames was performed using the MotionCor2 program. The CTF parameters of each aligned micrograph were estimated using the CTFFind4 program. Particle picking and two-dimensional classification were performed using the Relion software package. A total of 70,340 capsid particles were selected for further processing. To accelerate processing, the data were rescaled to the pixel size of 1.85 Å. The previously reported cryo-EM structure of the expanded T4 capsid[22], low-pass filtered to 40 Å resolution, was used as an initial model for determination of the alignment parameters of each particle and calculation of the initial 3D reconstruction using Relion. The 3D reconstruction was then refined using the Relion 3D autorefine procedure, assuming the D5 symmetry of the capsid shell. The refined map was post-processed in Relion and sharpened using the B-factor of −129 Å2. The resolution of the map was estimated to be 3.9 Å according to the gold standard Fourier shell correlation (FSC) cutoff of 0.143.

The atomic structure of the 9DE-T4 capsid was built with the help of the previously reported structure of the expanded T4 capsid (PDB ID: 7VS5)[22]. The gp23* and gp24* subunits were fitted into the cryo-EM density using the program Chimera. Then the structure was rebuilt manually using the program Coot and refined in the real space using the phenix.real_space_refine program. The refinement statistics are summarized in the Supplementary Table 2.

## Statistics and reproducibility
All quantified data are presented as the mean with SD. The measurements were taken from distinct or independent biological samples (n = 3−5 per individual group). The experiments were replicated at least three times, and the representative data, micrographs/images are

shown. Photoshop CS6 software was used to organize the figures. Statistical analyses were performed by ANOVA and $t$ tests as detailed in each section using GraphPad Prism 9 software. The difference between the two groups was considered statistically significant when $*p < 0.05$, or highly significant when $**p < 0.01$, $***p < 0.001$, and $****p < 0.0001$, or not significant (n.s.) when $p > 0.05$.

### Reporting summary
Further information on research design is available in the Nature Portfolio Reporting Summary linked to this article.

## Data availability
The authors declare that all data supporting the findings of this study are included in the manuscript and its Supplementary Information files. The cryo-EM reconstruction of 9DE-T4 capsid generated in this study has been deposited in the Electron Microscopy Data Bank under the accession code EMD-40228 and in the Protein Data Bank under the accession number PDB: 8GMO. Source data are provided with this paper.

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

## Acknowledgements

The authors thank Dr. Victor Padilla-Sanchez for assistance with the preparation of the supplementary movie 1 and figures, Dr. Qianglin Fang (Purdue University) for electron microscopy images, Drs. Thomas Klose and Jingchuan Sun for their help with 9DE-T4 cryo-EM data collection, and Drs. Mavis Agbandje-McKenna and Mario Mietzsch (University of Florida) for providing the AAV2 (Luci) virus. This research was supported by NIAID/NIH grant AI111538 and in part by the National Science Foundation grant MCB-0923873 and NIAID/NIH grants AI081726 and AI175340 to V.B.R.

## Author contributions

V.R. designed and directed the project. J.Z. and V.R. designed the research. J.Z., H.B., N.A., M.M., P.T., X.W., W.G., and A.F. performed the experiments. J.Z. and V.R. analyzed and interpreted the data. V.R. and J.Z. wrote the manuscript.

## Competing interests

The authors declare no competing interests.
