## [Peer Review File · Nature Communications]

Reviewers' Comments:

Reviewer #1:

Remarks to the Author:

Summary:

Zhu et al. present a very extensive study demonstrating the potential of heavily engineered T4-like phages as gene therapy vectors. They build on their extensive experience of using T4 as a vaccine candidate and employ an assembly-line like approach where DNA is packaged into empty phage heads and subsequently decorated with proteins and/or RNA via fusion to outer capsid proteins (hoc/soc). Uptake into mammalian cells is achieved through coating of these VLPs with cationic lipids.

The authors use this platform to deliver fluorescent and luminescent reporter genes or large therapeutic genes such as the 11 kb dystrophin. They also achieve co-delivery of DNA and proteins via hoc- and soc-fusion proteins on hoc/soc-deficient VLPs. In an impressive tour de force the authors demonstrate genome editing via delivery of Cas9/gRNA genes or via delivery of "loaded" Cas9/gRNA RNP complexes on the capsid (and combinations of both strategies). They achieve up to 35 % editing efficiency in human cell lines. Through capsid-mediated delivery of Cas9/gRNA RNP complexes combined with delivery of packaged editing templates, the authors also achieve site-specific recombination in various setups. In addition, they manage to deliver siRNA and mRNA by using Cas9 as a capsid-tethered RNA-binding protein, enabling target gene silencing (siRNA) or mRNA translation. To increase transduction efficiency, the authors introduce acidic amino acids into a solvent exposed loop in the capsid protein Gp23. The resulting super-acidic capsid surface leads to a 5-fold increase in transduction efficiency. Finally, the authors target an NLS-LacI to the capsid interior via a previously identified capsid-targeting sequence. NLS-LacI binds to packaged, lacO-containing DNA and boosts nuclear delivery of DNA cargo.

The manuscript is generally well written, easy to follow, and the evidence supports the conclusions.

I would like to sincerely congratulate the authors for an outstanding piece of work. This is the most impressive phage paper I have read in the past 5 years with therapeutic relevance far beyond phage or phage therapy. I would strongly support publication and have only few comments:

General comments:

- The authors should exchange the term "artificial virus" (AV) with "artificial virus-like particle". An AV would be an artificial structure than can replicate all stages of a virus infection, including multiplication, assembly, and egress. Since these modified phages cannot replicate and release progeny, the term AV is not correct.

-The abstract could be re-written/improved.

-The authors have a system that allows them to produce VLPs that modify the human genome. I would like to see a discussion on the safety of these vectors, both in their therapeutic use but also simply their handling in the lab. E.g., what would be a the required/suggested biosafetly level for working with agents that can edit your genome? How are the scientists protected?

Specific comments:

- L118: I would use the term "mimic an envelope" rather than "create an envelope", simply because it is quite different from an actual lipid envelope.

- Fig2B: EM control without lipid is missing

-L215-228: The elucidation of the AV entry pathway seems a bit weak (testing of 5 inhibitors, but no genetic assessment). Only 1 cell type. The effects are significant but weak, given that luminescence is typically measured in the log-scale (Fig. S2F). I would actually suggest to remove

the data and address this properly in a separate study. In my opinion, removing this data would not weaken the paper at all.

-L224: The contribution of endosomal acidification could be tested with inhibitors such as bafilomycin, chloroquine, or NH₄Cl.

Best regards,
Samuel Kilcher

Reviewer #2:

Remarks to the Author:

The authors provide an extensive report of their progress in leveraging bacteriophages for gene delivery to human cells, making an important contribution to biotechnology. They utilise the genetically tractable system of E.coli bacteriophage T4 to create mutant viruses capable of producing capsids with altered DNA and protein cargoes and modified external surfaces. The authors demonstrate that the virus-like particles can be taken up by HEK293T cells, through the use of reporter molecules such as fluorescent proteins.

General comments:

1) It should be mentioned in the introductory text that the approach of using bacteriophage T4 capsids with proteins targeted to their interior with the capsid targeting sequence (CTS), also utilising the Cre-Lox recombination system, with fusion of proteins to Hoc/Soc proteins, has been demonstrated before. The following paper should be cited in relation to this: Liu et al 2014. PNAS, 111(37):13319-24. doi: 10.1073/pnas.1321940111. The authors may then wish to emphasise how the present study builds upon this previous work, for example in the choice of molecules attached to the outside and inside of the capsid.

2) The term "artificial virus", used throughout, is not particularly accurate. What is being engineered from a virus is a virus-like particle (VLP) or, more broadly, the term the authors use on line 29: a nanoparticle. For the engineered particles to be considered as artificial viruses, or virions, they would need to target a host cell specifically using a receptor-targeting protein, and to then replicate inside the host cell to produce progeny capable of repeating this process. The term artificial virus can be replaced with a more appropriate term such as virus-like particle (VLP).

3) Data supporting electron microscopy is currently scant in results and methods. Figure 1 may be better used to display the reconstruction achieved in the present study (currently Figure 6), rather than models from previous work which already feature in subsequent figures. The authors report good resolution for 9DE-T4, certainly sufficient for the present analysis, and a model for the 9DE-T4 capsid appears to have been built (Figure 6 B). Measures or statistics related to this model should be provided. The methods state "structures of 9DE-gp23* and gp24* were reconstructed with the guide of the reported gp23 and gp24 crystal structures": more detail should be provided.

4) Have cryo-EM maps and models been deposited with the PDB or are these already publicly accessible. Figure 1 F and G should show a colour scale, as the exterior of the capsid is likely mixed in character, as is evident from Figure 6 B. Figure 2 B should be enlarged as it is difficult to see the capsids. Ideally further images of capsids in cryo-electron micrographs could be provided, but these could reside in supplemental data given the size of figures.

5) Clarity needs to be improved about structural diagrams (in particular on Fig.1): it is not always clear how each figure was generated – which PDB codes were used? Which diagrams are based on real structural data (cryo-EM or X-ray) and which are models not based on accurate structural information, and hence should be classified as schematics?

minor comments:

Line 47 – “can impair or kill a human person consisting of about 37 trillion cells”: The authors may wish to amend this sentence. The numbers of nucleotides and numbers of cells is fairly unremarkable without comparison, and bacterial viruses pose no or little risk to humans, which is the interest here.

Line 58 – “despite many attempts over the years”: Add references here.

Line 69 – Rephrase in light of comment above citing Liu et al 2014.

Reviewer #3:

Remarks to the Author:

The manuscript reports a process for constructing a phage-based gene delivery vector. The noteworthy results are the ability of the phage capsid to encase large DNA, RNA, and protein payload, as well as combinations of those, and to deliver the cargo into cells with an efficiency high enough to produce a readout.

I believe the work is of significance in the field of non-viral gene delivery and phage biology.

The corresponding author has an impressive publication record on the subject of genetic engineering of bacteriophages as vaccines or for gene delivery and the current manuscript is clearly built on years of experience. The authors have explained this well in the introduction.

I have read the paper carefully and made a list of the claims. The data very clearly show that the bacteriophage vector can effectively deliver a range of payloads. What is less clear is how this vector compares to the state-of-the-art. In other words, experiments seem to lack a positive control. This lack of a positive control for all experiments in general, and for gene delivery data in particular, is my main concern.

Other questions/concerns:

1-The abstract lacks quantitative data to backup the four claims stated, namely large capacity, custom buildable, multiplex, all-in-one. I understand that some are easier to quantify than others, but I know that most of these claims can be quantified (directly or indirectly) as has been done in the manuscript text.

2-Page 4, line 70: Phage T4 is from the Myoviridae family

3-Page 4, lines 71 and 72: please double check terminology used for scientific correctness. the reference cited does not seem to state these numbers.

4-flow of figure 1 is counterintuitive, and I would understand if this was the only way to present the information, but it's not. It can easily be rearranged to show A, then B, then C, and so on.

5-Please clarify if figure one schematics are produced by the authors from this manuscript based on their own data, or from other published work. The authors cite references in the text, so this seems to be from other published work. If so, this must be acknowledged accordingly in the figure caption

6-line 160, a TEM of virus with no lipids would be helpful. Since this claim (lipid coverage of virus capsid) is only backed up by qualitative data, the visuals matter in this case

7- Figure 2C presents another set of qualitative data to support the claim of lipid coverage. While visually appealing, these images do not support the claim stated in the text, namely high lipid coverage. One way to support this claim is to add different amounts of lipid and show the fluorescence intensity increase, which could also be quantified.

8-line 167: "The T4 artificial viruses efficiently deliver genetic payloads into human cells". The claim of efficiency must be backed up, preferably quantitatively

9-line 175: "these AVs successfully transduced both the reporter plasmids into HEK293T (293) cells at near 100% efficiency". How was this quantified?

10-Line 192, "The difference was greatest at low MOI (ratio of capsid particles to cells), which decreased with increasing MOI." Why? Why at low MOI's? same on line 228: why does the low MOI work better than high MOI?

11- The Introduction is very lengthy, 3.5 pages. I suggest the authors summarize the information and stick to a tight storyline.

Q:Are there any flaws in the data analysis, interpretation and conclusions?

A:In most cases, the term "efficiency" is used without offering a concrete definition. This will prohibit repeatably. I strongly suggest this term be defined for each case.

Q:Do these prohibit publication or require revision?

A: No, I suggest a revised version be considered for publication.

Q: Is the methodology sound? Does the work meet the expected standards in your field?

A: yes and yes, see comment about positive control.

Q: Is there enough detail provided in the methods for the work to be reproduced?

A; to the best of my knowledge, yes.

RESPONSES TO REVIEWERS' COMMENTS

Reviewer #1 (Remarks to the Author):

Summary:

Zhu et al. present a very extensive study demonstrating the potential of heavily engineered T4-like phages as gene therapy vectors. They build on their extensive experience of using T4 as a vaccine candidate and employ an assembly-line like approach where DNA is packaged into empty phage heads and subsequently decorated with proteins and/or RNA via fusion to outer capsid proteins (hoc/soc). Uptake into mammalian cells is achieved through coating of these VLPs with cationic lipids.

The authors use this platform to deliver fluorescent and luminescent reporter genes or large therapeutic genes such as the 11 kb dystrophin. They also achieve co-delivery of DNA and proteins via hoc- and soc-fusion proteins on hoc/soc-deficient VLPs. In an impressive tour de force the authors demonstrate genome editing via delivery of Cas9/gRNA genes or via delivery of "loaded" Cas9/gRNA RNP complexes on the capsid (and combinations of both strategies). They achieve up to 35 % editing efficiency in human cell lines. Through capsid-mediated delivery of Cas9/gRNA RNP complexes combined with delivery of packaged editing templates, the authors also achieve site-specific recombination in various setups. In addition, they manage to deliver siRNA and mRNA by using Cas9 as a capsid-tethered RNA-binding protein, enabling target gene silencing (siRNA) or mRNA translation. To increase transduction efficiency, the authors introduce acidic amino acids into a solvent exposed loop in the capsid protein Gp23. The resulting super-acidic capsid surface leads to a 5-fold increase in transduction efficiency. Finally, the authors target an NLS-LacI to the capsid interior via a previously identified capsid-targeting sequence. NLS-LacI binds to packaged, lacO-containing DNA and boosts nuclear delivery of DNA cargo.

The manuscript is generally well written, easy to follow, and the evidence supports the conclusions.

I would like to sincerely congratulate the authors for an outstanding piece of work. This is the most impressive phage paper I have read in the past 5 years with therapeutic relevance far beyond phage or phage therapy. I would strongly support publication and have only few comments:

We thank the reviewer for the kind words and the insightful comments. We are also grateful for the reviewer's recognition of the significance and therapeutic potential of our work, which inspires us.

General comments:

- The authors should exchange the term “artificial virus” (AV) with “artificial virus-like particle”. An AV would be an artificial structure than can replicate all stages of a virus infection, including multiplication, assembly, and egress. Since these modified phages cannot replicate and release progeny, the term AV is not correct.

We agree. In fact, we have gone through several options including artificial “virus-like particle” before settling on artificial “virus”. Though the T4-AVs cannot replicate or produce progeny, they do mimic the typical architecture of engineered eukaryotic viruses used in biomedical applications, which are also replication deficient and cannot produce progeny¹. For example, in most widely used AAV vectors, the viral genes are often completely removed, leaving only the inverted terminal repeats (ITRs).

However, in light of the comment, we take the reviewer’s suggestion and replace “artificial virus” with “artificial viral vector” (AVV), which might be a more appropriate term because it is used as a vehicle to deliver biomolecules into cells. We believe that this term better reflects the focus and purpose of our research.

The title and text throughout the manuscript are changed accordingly.

-The abstract could be re-written/improved.

We have carefully gone through the abstract that has a limit of 150 words and made several changes to improve it.

-The authors have a system that allows them to produce VLPs that modify the human genome. I would like to see a discussion on the safety of these vectors, both in their therapeutic use but also simply their handling in the lab. E.g., what would be a required/suggested biosafety level for working with agents that can edit your genome? How are the scientists protected?

There is no reasonable concern regarding the safety of the T4-AVVs because they lack the ability to replicate. However, we perform T4-AVV assembly and transduction under standard laboratory conditions that are equivalent to biosafety level 2. These conditions include safety protocols such as the use of appropriate personal protective equipment and maintaining aseptic conditions to minimize the risk of contamination. Similar safety measures are also used for eukaryotic viral vectors such as AAV and lentiviruses that are widely used in animal research and human gene therapy trials.

To the reviewer’s point, safety concerns may arise when T4-AVVs are transitioned to clinic, such as potential unwanted responses by the host immune

system or off-target effects. These will be examined and appropriately addressed as we make progress towards the clinic.

A statement regarding the safety is added to Discussion and Methods (lines 623-625 and 762-765).

Specific comments:

- L118: I would use the term “mimic an envelope” rather than “create an envelope”, simply because it is quite different from an actual lipid envelope. Changed as suggested (line 106).

- Fig2B: EM control without lipid is missing

As suggested, we have provided additional negative EM images of T4 heads with and without lipid. Additionally, in response to a similar comment by Reviewer #3, we have included quantitative data for binding of lipid to T4 head particles. Because of space considerations, these data are shown in Supplemental Fig. 2a-2c, in the revised manuscript.

-L215-228: The elucidation of the AV entry pathway seems a bit weak (testing of 5 inhibitors, but no genetic assessment). Only 1 cell type. The effects are significant but weak, given that luminescence is typically measured in the log-scale (Fig. S2F). I would actually suggest to remove the data and address this properly in a separate study. In my opinion, removing this data would not weaken the paper at all.

As suggested, the data in S2F are removed. We agree that this would not affect the paper.

However, a sentence referring to these observations is needed to provide context for the section “Re-wiring the capsid exterior” (lines 436-438). Additionally, this is also mentioned in Discussion as a question that requires detailed investigations in the future (lines 611-613, see below).

-L224: The contribution of endosomal acidification could be tested with inhibitors such as bafilomycin, chloroquine, or NH₄Cl.

As per the reviewer’s suggestion, we conducted an experiment to assess the role of endosomal acidification in T4-AVV delivery efficiency by using the inhibitor chloroquine. Despite its well established enhancement of endosomal escape through the “proton sponge” effect², our results showed no significant improvement in delivery efficiencies by T4-AVVs, as measured by luciferase expression at 75 and 150 μM chloroquine. Therefore, it appears, that the T4-AVVs may not rely heavily on this mechanism for endosome escape. However, as stated in lines 611-613, this requires a deeper investigation of endosomal entry and escape pathways, which we plan to investigate in future in a separate study as suggested by the reviewer.

Best regards,
Samuel Kilcher

Reviewer #2 (Remarks to the Author):

The authors provide an extensive report of their progress in leveraging bacteriophages for gene delivery to human cells, making an important contribution to biotechnology. They utilise the genetically tractable system of E.coli bacteriophage T4 to create mutant viruses capable of producing capsids with altered DNA and protein cargoes and modified external surfaces. The authors demonstrate that the virus-like particles can be taken up by HEK293T cells, through the use of reporter molecules such as fluorescent proteins.

We thank the reviewer for the positive and insightful comments.

General comments:

1) It should be mentioned in the introductory text that the approach of using bacteriophage T4 capsids with proteins targeted to their interior with the capsid targeting sequence (CTS), also utilising the Cre-Lox recombination system, with fusion of proteins to Hoc/Soc proteins, has been demonstrated before. The following paper should be cited in relation to this: Liu et al 2014. PNAS, 111(37):13319-24. doi: 10.1073/pnas.1321940111. The authors may then wish to emphasise how the present study builds upon this previous work, for example in the choice of molecules attached to the outside and inside of the capsid.

We have included the above reference and also modified the text in Introduction and Results to further connect the present studies to the previously published reports on the development of protein packaging platform by Black and coworkers (lines 95-96 and 491-494).

2) The term "artificial virus", used throughout, is not particularly accurate. What is being engineered from a virus is a virus-like particle (VLP) or, more broadly, the term the authors use on line 29: a nanoparticle. For the engineered particles to be considered as artificial viruses, or virions, they would need to target a host cell specifically using a receptor-targeting protein, and to then replicate inside the host cell to produce progeny capable of repeating this process. The term artificial virus can be replaced with a more appropriate term such as virus-like particle (VLP).

We agree. In fact, as stated in response to a similar comment above by Reviewer #1, we have gone through several options including "virus-like particle" before settling on the term artificial "virus". Although the T4-AVs cannot replicate or produce progeny, they do mimic the typical architecture of

engineered eukaryotic viruses that too are replication deficient and cannot produce progeny¹.

However, in light of the comments, we take the reviewers' suggestion and replace "artificial virus" with "artificial viral vector" (AVV), which might be a more appropriate term because the assembled T4 is used as a vehicle to deliver biomolecules into cells, similar to the widely used viral vectors, AAV and lentivirus. We believe that this term better reflects the focus and purpose of our research.

The title and text throughout are changed accordingly.

3) Data supporting electron microscopy is currently scant in results and methods. Figure 1 may be better used to display the reconstruction achieved in the present study (currently Figure 6), rather than models from previous work which already feature in subsequent figures. The authors report good resolution for 9DE-T4, certainly sufficient for the present analysis, and a model for the 9DE-T4 capsid appears to have been built (Figure 6 B). Measures or statistics related to this model should be provided. The methods state "structures of 9DE-gp23* and gp24* were reconstructed with the guide of the reported gp23 and gp24 crystal structures": more detail should be provided.

Figure 1 may be better used to display the reconstruction achieved in the present study (currently Figure 6), rather than models from previous work which already feature in subsequent figures. The authors report good resolution for 9DE-T4, certainly sufficient for the present analysis, and a model for the 9DE-T4 capsid appears to have been built (Figure 6 B).

While we agree that the cryo-EM structure of the 9DE-T4 head is resolved to a high resolution (3.9 Å), this structure does not include Hoc and Soc components. Therefore, the wild-type T4 head structural model in Fig. 1 that includes both Hoc and Soc better reflects the studies described.

As suggested by the reviewer, we have also included the cryo-EM structures (surface views) of both wild-type T4 (3.4 Å)³ (Fig. 1f) and 9DE-T4 (3.9 Å) (Fig. 1g) heads for comparison of surface charges. Additionally, the figure legend is clarified by including the sources of the structural models and the relevant PDB codes and references (see below).

Measures or statistics related to this model should be provided. The methods state "structures of 9DE-gp23 and gp24* were reconstructed with the guide of the reported gp23 and gp24 crystal structures": more detail should be provided.*

As suggested by the reviewer, we have added detailed descriptions of the Cryo-EM imaging and data processing in the Methods section (lines 977-1003).

Additionally, the measures and statistics are provided in Supplemental table 2 in the revised manuscript.

4) Have cryo-EM maps and models been deposited with the PDB or are these already publicly accessible. Figure 1 F and G should show a colour scale, as the exterior of the capsid is likely mixed in character, as is evident from Figure 6 B. Figure 2 B should be enlarged as it is difficult to see the capsids. Ideally further images of capsids in cryo-electron micrographs could be provided, but these could reside in supplemental data given the size of figures.

Have cryo-EM maps and models been deposited with the PDB or are these already publicly accessible.

The 3.4 Å resolution cryo-EM map of the wild-type T4 head has been deposited in the PDB with accession number 7VS5³. We have included this and other accession numbers in the figure legend (Fig. 1). We are in the process of depositing the 9DE-T4 structure in the PDB and will provide this information once it is approved.

Figure 1 F and G should show a colour scale, as the exterior of the capsid is likely mixed in character, as is evident from Figure 6 B.

As suggested by the reviewer, we have included the cryo-EM structures (surface views) of both wild-type T4 (3.4 Å)³ and 9DE-T4 (3.9 Å) for comparison with a color scale in Fig. 1f and 1g, respectively.

Figure 2 B should be enlarged as it is difficult to see the capsids. Ideally further images of capsids in cryo-electron micrographs could be provided, but these could reside in supplemental data given the size of figures.

As suggested, Figure 2B is enlarged and also, additional negative EM images of T4 heads with and without lipid have been provided in the revised manuscript. Additionally, in response to a similar comment by Reviewer #3, we have included quantitative data for binding of lipid to T4 head particles. Because of space considerations, these data are shown in Supplemental Fig. 2a-2c, in the revised manuscript.

With regard to cryo-EM structure of T4-AVVs, it requires a detailed investigation in the future, including series of optimizations to clearly visualize the lipid coat.

5) Clarity needs to improved about structural diagrams (in particular on Fig.1): it is not always clear how each figure was generated – which PDB codes were used? Which diagrams are based on real structural data (cryo-EM or X-ray) and which are models not based on accurate structural information, and hence should be classified as schematics?

As suggested, we have updated the figure captions to include the accession numbers of all the T4 capsid and/or Hoc/Soc structures depicted in the figure.

Additionally, we have clearly designated in the figure legends which diagrams are based on the structural data obtained through cryo-EM or X-ray crystallography, and which are models (lines 1233-1245, 1248, 1280, 1284, 1310, 1349, 1377, 1379, 1382, and 1396).

Minor comments:

Line 47 – “can impair or kill a human person consisting of about 37 trillion cells”: The authors may wish to amend this sentence. The numbers of nucleotides and numbers of cells is fairly unremarkable without comparison, and bacterial viruses pose no or little risk to humans, which is the interest here.

As suggested, the statement is re-phrased as follows: “Despite their small size and simple genetic makeup, viruses can cause deadly infections or global pandemics such as AIDS, Flu, and COVID-19.” (Lines 46-47).

Line 58 – “despite many attempts over the years”: Add references here.

As suggested, the relevant references are added (lines 53-54).

Line 69 – Rephrase in light of comment above citing Liu et al 2014.

Modified as suggested (lines 95-96 and 491-494).

Reviewer #3 (Remarks to the Author):

The manuscript reports a process for constructing a phage-based gene delivery vector. The noteworthy results are the ability of the phage capsid to encase large DNA, RNA, and protein payload, as well as combinations of those, and to deliver the cargo into cells with an efficiency high enough to produce a readout. I believe the work is of significance in the field of non-viral gene delivery and phage biology.

The corresponding author has an impressive publication record on the subject of genetic engineering of bacteriophages as vaccines or for gene delivery and the current manuscript is clearly built on years of experience. The authors have explained this well in the introduction.

We thank the reviewer for the positive and insightful comments.

I have read the paper carefully and made a list of the claims. The data very clearly show that the bacteriophage vector can effectively deliver a range of payloads. What is less clear is how this vector compares to the state-of-the-art. In other words, experiments seem to lack a positive control. This lack of a positive control for all experiments in general, and for gene delivery data in particular, is my main concern.

We appreciate the reviewer for these comments.

Regarding the concern on the positive controls, we did have controls for gene delivery experiments. For comparison of the delivery efficiency by T4-AVVs with a positive control, we used AAV2, a widely used eukaryotic viral vector for efficient gene delivery (Fig. 2e). Additionally, we have also included lipofectamine liposome control for gene delivery and genome editing (Fig. 4f). When determining the transduction efficiency of various modified T4-AVVs (protein/RNA display, exterior wiring and interior wiring), we also used unmodified T4-AVV as a control (Fig. 3-7).

For delivery of large and complex payloads, such as large DNAs, multiple genes and proteins, or protein-DNA-RNA complexes, there is currently no equivalent positive control we could use. Delivering such large payloads is a unique property of the T4 platform that has not been achieved by traditional vectors such as AAV (~5 Kbp max), lentivirus (~8 Kbp max), or nanoparticle systems due to size and engineering limitations⁴. In these experiments, we used an internal control, delivery and expression of a reporter gene such as *luciferase*, *GFP*, or *mCherry* co-packaged into the same head, to assess the transduction efficiency.

These points are further clarified in our Response to Comment #11 below. We have carefully gone through the paper and included appropriate definitions or comparison to qualify the stated efficiency (lines 156-160, 173-176, 227-231, 294-296, 307-311, 346-347, 360-361, 389-390, 392-394, 463-467, 502-505, and 783-787). We have also deleted or modified statements where the use of the word “efficient” might be ambiguous (lines 152-153, 194, 243, 259, 352, 359, 387, 390, 401, 520-521, 1340, and 1403).

Other questions/concerns:

1-The abstract lacks quantitative data to backup the four claims stated, namely large capacity, custom buildable, multiplex, all-in-one. I understand that some are easier to quantify than others, but I know that most of these claims can be quantified (directly or indirectly) as has been done in the manuscript text.

The 150-word limitation constrains us to limit the abstract to a broad summary of the work and does not allow inclusion of the quantitative data. We have gone through the abstract carefully and made several changes to improve it.

2-Page 4, line 70: Phage T4 if from the Myoviridea family.

Phage T4 has been recently reclassified as a member of the *Straboviridae* family, *Tevenvirinae* subfamily, and *Tequatrovirus* genus, according to the recent taxonomic updates (2021) (https://ictv.global/taxonomy/taxondetails?taxnode_id=202100332). This is a change from its previous classification as a member of the *Myoviridae* family. This reference is now added to the text (ref 17 in the text, line 63).

3-Page 4, lines 71 and 72: please double check terminology used for scientific correctness. The reference cited does not seem to state these numbers.

The statement is modified appropriately as follows: "With an infection efficiency near 100%⁵, and replicating at a rate of ~20-30 minutes per cycle⁶, T4 is one of the most efficient viruses known" (lines 63-64).

4-flow of figure 1 is counterintuitive, and I would understand if this was the only way to present the information, but it's not. It can easily be rearranged to show A, then B, then C, and so on.

Corrected as suggested.

5-Please clarify if figure one schematics are produced by the authors from this manuscript based on their own data, or from other published work. The authors cite references in the text, so this seems to be from other published work. If so, this must be acknowledged accordingly in the figure caption.

As suggested, we have clarified the figure legend by stating the sources of the structures/schematics derived from our previous studies and also acknowledging the respective references.

6-line 160, a TEM of virus with no lipids would be helpful. Since this claim (lipid coverage of virus capsid) is only backed up by qualitative data, the visuals matter in this case.

As per the reviewer's suggestion, we have provided additional negative EM images of T4 heads with and without lipid and included these images in Supplemental Fig. 2a.

7- Figure 2C presents another set of qualitative data to support the claim of lipid coverage. While visually appealing, these images do not support the claim stated in the text, namely high lipid coverage. One way to support this claim is to add different amounts of lipid and show the fluorescence intensity increase, which could also be quantified.

As per the reviewer's suggestion, we have included quantitative data for binding of lipid to T4 head particles in Supplemental Fig. 2b and 2c of the revised manuscript.

8-line 167: "The T4 artificial viruses efficiently deliver genetic payloads into human cells". The claim of efficiency must be backed up, preferably quantitatively.

As suggested, the following statement is added to further clarify the efficiency (lines 156-160): "when co-packaged with two different linear plasmids, on average ~5 molecules each of *GFP* reporter plasmid (5.4 Kbp) and *Luciferase* plasmid (*Luci*, 6.3 Kbp) per nanoparticle (Supplemental Fig. 1c), these AVVs were able to transduce the reporter plasmids into HEK293T (293) cells with

near 100% efficiency, as determined by the percentage of cells expressing GFP fluorescence (Fig. 2c and 2d)” (please also see below our response to comment #9).

9-line 175: "these AVs successfully transduced both the reporter plasmids into HEK293T (293) cells at near 100% efficiency". How was this quantified?

To quantify the efficiency of transduction, we counterstained the AAV-transduced cells with Hoechst (nucleus staining). We then counted the number of Hoechst-stained and GFP-Hoechst double-stained cells in three fields of microscope to approximately determine the efficiency of transduction. This was calculated as the number of GFP and Hoechst double-stained cells divided by the number of Hoechst-stained cells (or percentage of cells expressing GFP fluorescence). This information is now added to the text (lines 158-160 and 783-787).

10-Line 192, “The difference was greatest at low MOI (ratio of capsid particles to cells), which decreased with increasing MOI.” Why? Why at low MOI's? same on line 228: why does the low MOI work better than high MOI?

It is not that the low MOI works better, but it is the ratio of luciferase activity that does not increase proportionally with the MOI.

Line 192: At low MOIs, the ratio of T4-AVV/AAV luciferase activity is significantly higher than at high MOIs. As stated in the text (lines 176-178), this might be due to T4-AVV's ability to package and deliver ~8 molecules of *Luci* plasmid into a single cell during one transduction event, whereas AAV can only deliver one copy at a time. Consequently, at low MOIs, there would be increased number of transduced *Luci* molecules by T4-AVV transduction than by the AAV transduction and hence, increased luciferase activity.

At a high MOI, T4-AVVs may have reached their maximum transduction and expression capacity within the cell, causing the luciferase signal not to increase proportionally (Fig. 2e and Supplemental Fig. 2f). On the other hand, AAV transduction continues to increase proportionally with increasing MOI, thereby reducing the fold-difference in the ratio of luciferase activity between T4-AAV and AAV transductions (although the luciferase signal is still higher for T4-AVVs than for AAV transduction).

We modified our statements to make the above points clearer (Lines 173-183).

Line 228, TBA treatment: The T4-transduced DNAs appear to be more efficiently transported into the nucleus in the presence of TBA, which is reported to stabilize microtubules and enhance DNA transport from the cytoplasm to the nucleus. At low MOI, because of low numbers of transduced DNA, the TBA enhancement is more pronounced than at high MOI where maximum

transduction and signal capacity is reached and hence the increase is not proportional to increase in MOI.

As suggested by Reviewer 1, we have deleted this paragraph and mentioned it in Results (lines 436-438) and Discussion (lines 611-613) to provide appropriate context. More detailed investigations on the endosomal entry and escape, and intracellular DNA transport will be conducted in the future to elucidate the mechanisms.

11- The Introduction is very lengthy, 3.5 pages. I suggest the authors summarize the information and stick to a tight storyline.

As suggested, we made several changes to tighten the Introduction and reduced the length to 3 pages in the revised manuscript.

Q: Are there any flaws in the data analysis, interpretation and conclusions?

A: In most cases, the term "efficiency" is used without offering a concrete definition. This will prohibit repeatably. I strongly suggest this term be defined for each case.

We define efficiency as the ability of the T4-AVVs to deliver cargos into human cells, as assessed by a reporter gene expression signal such as the luciferase activity or GFP/mCherry fluorescence, or a functional assay of the delivered cargo such as the T7 *indel* nuclease assay, using various internal and/or external controls.

Some examples are as follows:

1) We demonstrated that T4-AVVs effectively transduced HEK293T cells by delivering reporter plasmids with near 100% efficiency under the conditions described, as measured by the percentage of cells expressing GFP fluorescence (Fig. 2c; see above under Comment #9). Additionally, we used transduction by AAV2, a well-established viral vector for efficient gene delivery, as an external control. Transduction efficiencies were compared on a head particle-to-head particle basis, each packaged with the same-sequence *ITR-Luci* plasmid. Our results showed that T4-AVVs exhibited 4-19 fold higher luciferase expression compared to AAV2 (Fig. 2e). These findings provide quantitative support for our claim of efficient delivery by T4-AVV without any exterior or interior modifications.

2) We assembled a series of T4-AVVs by i) displaying a variety of proteins (Fig. 3), ii) rewiring the exterior with acidic amino acids (Fig. 6), or iii) rewiring the interior by packaging foreign proteins (Fig. 7). In these experiments, there is an internal control in each experiment, packaging of one of the reporter genes into the same T4-AVVs and comparing transduction efficiency with that of the unmodified T4-AVVs described above. The modified T4-AVVs demonstrated even higher transduction efficiency compared to unmodified T4-AVVs based on

the packaged *Luciferase* or *Cas9-gRNA* gene expression, with a 2-4 fold increase (Fig. 3d, 6f, 7d, and 7e).

3) For genome editing T4-AVVs, we used both an internal control for transduction efficiency, i.e., reporter gene expression delivered by the same T4-AVVs (Fig 4e and Supplemental Fig. 4e and 4h), and also an external control, lipofectamine transfection that delivered the same genome editing genes. We report that the most efficient genome editing T4-AVVs showed ~30-35% indels at the AAVS1 locus, about twice that obtained by the lipofectamine control (Fig. 4f and Supplemental Fig. 4i).

4) For the gene-silencing T4-AVVs in which the gRNA is replaced with siRNA, we report nearly 100% silencing in 72 hrs was observed, based on Western blotting using a GFP monoclonal antibody (Fig. 4d).

It should, however, be noted that while we used a combination of internal and/or external controls as above to determine/control efficiencies, there is no equivalent external control we could use for the delivery of large and complex payloads such as large DNAs, multiple genes and proteins, or protein-DNA-RNA complexes because these are unique to the T4-AVV platform. None of the current vectors such as AAV (~5 Kbp max), or lentivirus (~8 Kbp max), or nanoparticle systems could deliver large and complex payloads due to size and engineering limitations.

We have carefully gone through the paper and included appropriate definitions or comparison to qualify the stated efficiency (lines 156-160, 173-176, 227-231, 294-296, 307-311, 346-347, 360-361, 389-390, 392-394, 463-467, 502-505, and 783-787). We have also deleted or modified statements where the use of the word "efficient" might be ambiguous (lines 152-153, 194, 243, 259, 352, 359, 387, 390, 401, 520-521, 1340, and 1403).

Q: Do these prohibit publication or require revision?

A: No, I suggest a revised version be considered for publication.

Q: Is the methodology sound? Does the work meet the expected standards in your field?

A: yes and yes, see comment about positive control.

Q: Is there enough detail provided in the methods for the work to be reproduced?

A; to the best of my knowledge, yes.

We thank the reviewer for these overall positive conclusions about our studies.

References

- 1 Bulcha, J. T., Wang, Y., Ma, H., Tai, P. W. L. & Gao, G. Viral vector platforms within the gene therapy landscape. *Signal Transduct. Target. Ther.* **6**, 53 (2021).
- 2 Seglen, P. O., Grinde, B. & Solheim, A. E. Inhibition of the lysosomal pathway of protein

- degradation in isolated rat hepatocytes by ammonia, methylamine, chloroquine and leupeptin. *FEBS J.* **95**, 215-225 (1979).
- 3 Fang, Q. *et al.* Structures of a large prolate virus capsid in unexpanded and expanded states generate insights into the icosahedral virus assembly. *Proc. Natl. Acad. Sci. U.S.A.* **119**, e2203272119 (2022).
- 4 Raguram, A., Banskota, S. & Liu, D. R. Therapeutic in vivo delivery of gene editing agents. *Cell* **185**, 2806-2827 (2022).
- 5 Storms, Z. J., Teel, M. R., Mercurio, K. & Sauvageau, D. The Virulence Index: A Metric for Quantitative Analysis of Phage Virulence. *PHAGE* **1**, 27-36 (2020).
- 6 Storms, Z. J., Brown, T., Cooper, D. G., Sauvageau, D. & Leask, R. L. Impact of the cell life-cycle on bacteriophage T4 infection. *FEMS Microbiol. Lett.* **353**, 63-68 (2014).

Reviewers' Comments:

Reviewer #1:

Remarks to the Author:

I would like to thank the authors for addressing my comments in a satisfactory manner. I have no further comments. In my opinion, the manuscript is ready for publication.

Reviewer #2:

Remarks to the Author:

My comments have been adequately addressed, and I am happy for this manuscript to be published.

Reviewer #3:

Remarks to the Author:

All my concerns have been addressed. I'm happy with the response letter and the revisions made to the text and figures. I suggest the manuscript to be published.

REVIEWERS' COMMENTS

Reviewer #1 (Remarks to the Author):

I would like to thank the authors for addressing my comments in a satisfactory manner. I have no further comments. In my opinion, the manuscript is ready for publication.

Reviewer #2 (Remarks to the Author):

My comments have been adequately addressed, and I am happy for this manuscript to be published.

Reviewer #3 (Remarks to the Author):

All my concerns have been addressed. I'm happy with the response letter and the revisions made to the text and figures. I suggest the manuscript to be published.

We appreciate all the reviewers for the positive and insightful comments.